# Test-Time Mixture of World Models for Embodied Agents in Dynamic Environments

**Jinwoo Jang, Minjong Yoo, Sihyung Yoon & Honguk Woo**[*]
Department of Computer Science and Engineering
Sungkyunkwan University
Suwon, Republic of Korea
{jinustar, mjyoo2, godboy3752, hwoo}@skku.edu

## Abstract

Language model (LM)-based embodied agents are increasingly deployed in real-world settings. Yet, their adaptability remains limited in dynamic environments, where constructing accurate and flexible world models is crucial for effective reasoning and decision-making. To address this challenge, we extend the Mixture-of-Experts (MoE) paradigm to embodied agents. While conventional MoE architectures modularize knowledge into expert components with pre-trained routing, they remain rigid once deployed, making them less effective for adapting to unseen domains in dynamic environments. We therefore propose Test-time Mixture of World Models (TMoW), a framework that enhances adaptability to unseen and evolving domains. TMoW updates its routing function over world models at test time, unlike conventional MoE where the function remains fixed, enabling agents to recombine existing models and integrate new ones for continual adaptation. It achieves this through (i) multi-granular prototype-based routing, which adapts mixtures across object- to scene-level similarities, (ii) test-time refinement that aligns unseen domain features with prototypes during inference, and (iii) distilled mixture-based augmentation, which efficiently constructs new models from few-shot data and existing prototypes. We evaluate TMoW on Virtual-Home, ALFWorld, and RLBench benchmarks, demonstrating strong performance in both zero-shot adaptation and few-shot expansion scenarios, and showing that it enables embodied agents to operate effectively in dynamic environments.

## 1 Introduction

Embodied agents powered by language models (LMs) are increasingly deployed in diverse environments such as households (Song et al., 2023; Ahn et al., 2022), factories (Kang et al., 2024; Zhang et al., 2023), and virtual games (Zhao et al., 2024; Fan et al., 2022). However, their monolithic architectures restrict adaptation, with capabilities frozen at training and domain knowledge buried in billions of shared parameters. This is especially problematic for embodied agents in real-world settings, where tasks and environments change constantly beyond training. In practice, this lack of adaptability leaves retraining as the only option, which imposes significant computational and data collection burdens and hinders real-world deployment. While in-context learning (Song et al., 2023; Kim et al., 2025) attempts to address this through domain-specific prompting, it merely shifts computational burden to inference with inflated context windows, highlighting the need for truly modular architectures.

Mixture-of-Experts (MoE) (Jacobs et al., 1991) architectures provide structural modularity by selectively activating expert modules per input, thereby supporting scalable inference and domain specialization. Building on this property, recent work has applied MoE to domain adaptation in LM-based agents through techniques such as meta-distillation (Zhong et al., 2022), vision-language models (Iftee et al., 2024), and anomaly detection (Lei et al., 2024).

In this work, we investigate MoE architectures for LM-based embodied agents, aiming to support rapid adaptation to unseen domains in dynamic environments. While MoE can flexibly activate

---

[*]Corresponding author

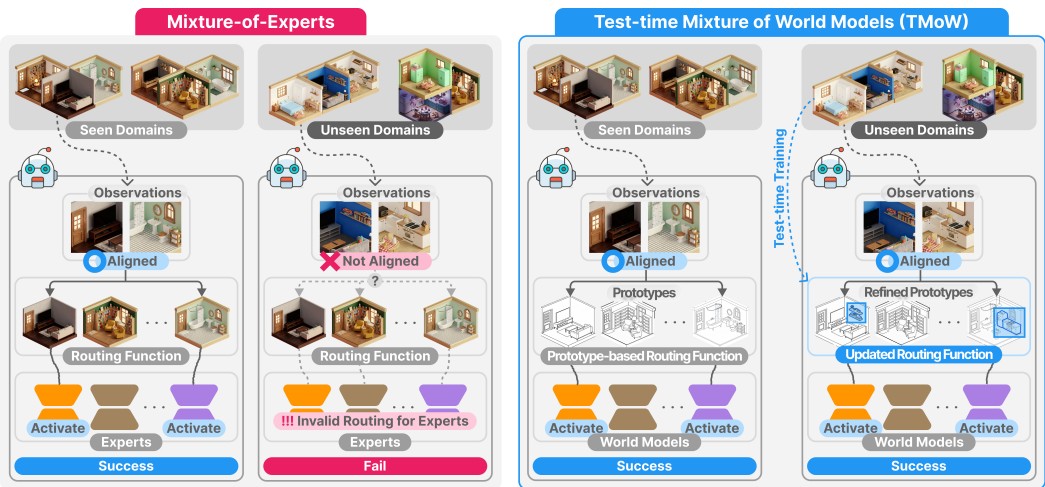

Figure 1: Existing Mixture-of-Experts employs a fixed router after training, making adaptation to unseen domains costly and requiring retraining (left). In contrast, **Test-time Mixture of World Models (TMoW)** overcomes this through prototype-based routing and test-time training of the routing function to reflect new domain characteristics without demonstrations (right).

domain-specific experts, their routing functions, responsible for selecting appropriate expert modules for each input, remain fixed after training, making adaptation to novel domains possible only through costly retraining. For embodied agents in temporally and spatially changing environments, it is essential to overcome this rigidity by adaptively reconfiguring routing functions and extending domain experts to meet these evolving conditions.

To address this need, we present a Test-time Mixture of World Models (TMoW) framework (Figure 1), which performs test-time training of a routing function without requiring demonstrations. This test-time training approach enables rapid adaptation to environments with temporal and spatial variations beyond the training distribution, facilitating continuous expansion of the model's capabilities. In the framework, world models act as internal simulators that allow the agent to predict environmental dynamics, reason about future outcomes, and plan appropriate actions, where each model corresponds to a distinct domain within the environment (Zhu et al., 2025; Janner et al., 2021).

TMoW employs a **multi-granular prototype-based router** that adapts world model mixtures by comparing input observations with learned prototype representations across different levels of spatial abstraction, ranging from local objects to global scenes. We adopt this hierarchical design inspired by LMs' proven ability to build effective representations through layer-wise progression from local tokens to global context (Van Aken et al., 2019). This router enables **test-time prototype refinement** to unseen domains by refining prototypes through weighted interpolation between existing prototypes based on their similarity to the current environment. Furthermore, TMoW supports **distilled mixture-based model augmentation** for data-efficient creation of new world models. Unlike test-time prototype refinement, this technique requires few-shot demonstrations and distills knowledge from existing model mixtures to construct new models for unseen domains, thereby incrementally expanding TMoW's repertoire after deployment.

To evaluate TMoW, we conduct experiments on VirtualHome (Puig et al., 2018), ALFWorld (Shridhar et al., 2021), RLBench (James et al., 2020), and real-world robotic scenarios, demonstrating its capability for rapid and scalable adaptation for evolving and expanding environments. Our framework achieves a 27.21% improvement over SayCanPay (Hazra et al., 2024), state-of-the-art baselines in zero-shot adaptation, and a 25.66% gain in few-shot expansion scenarios when constructing new world models.

Our contributions are as follows. (1) We propose the TMoW framework, a novel extension of MoE that supports test-time reconfiguration of expert mixtures (i.e., test-time mixture), allowing embodied agents to adapt to unseen domains without costly retraining. (2) We develop a multi-granular prototype-based routing mechanism that realizes this test-time mixture capability, leveraging prototype similarity across spatial levels ranging from local objects to global scenes. (3) We devise

a distilled mixture-based model augmentation strategy that expands TMoW's adaptability by data-efficiently constructing new world models. (4) We validate TMoW on diverse benchmarks and real-world scenarios, demonstrating that it advances MoE-based adaptation and offers an effective solution for embodied agents operating in evolving physical environments.

## 2    RELATED WORK

**LM-based embodied instruction following.**    Embodied instruction following requires agents to ground language instructions in physical environments through sequential action execution, which in turn necessitates robust world models capable of representing diverse environmental configurations. Several approaches have been introduced. Code-driven policies (Singh et al., 2023; Liang et al., 2023) generate executable programs from instructions, while reward-based methods (Yu et al., 2023; Adeniji et al., 2023) learn value functions to guide action selection. Furthermore, LM-based reasoning has been combined with domain-specific models to assess environmental affordances (Ahn et al., 2022; Hazra et al., 2024), and in-context learning approaches incorporate relevant demonstrations directly into the agent's decision-making process (Song et al., 2023). However, these existing approaches often struggle to generalize to new environments without significant retraining or increase inference-time overhead. Our TMoW framework addresses these challenges by introducing test-time adaptation directly into the MoE architecture, enabling efficient adjustment to unseen domains without retraining.

**Mixture-of-Experts.**    The Mixture-of-Experts (MoE) provides efficient model scaling by activating only a subset of experts, as shown in sparse MoE transformers like GShard (Lepikhin et al., 2020) and Switch Transformer (Fedus et al., 2022). MoE has also been applied to various adaptation settings including Meta-DMoE (Zhong et al., 2022) which uses meta-distillation from domain-specific experts, and MoE-TTA (Iftee et al., 2024) which supports adaptation for vision-language models. Yet, existing MoE architectures face limitations in knowledge expansion, as experts are tightly coupled within a single training graph. Incorporating new domains typically requires end-to-end retraining or costly knowledge distillation, thereby restricting modular extensibility. This rigid design prevents modular growth, making it unsuitable for evolving physical environments with diverse tasks and domains. Our approach addresses this through prototype-based routing, which directly supports rapid integration of new world models and dynamic mixture adaptation, while also facilitating knowledge expansion through few-shot model augmentation.

**Test-time adaptation.**    Test-time adaptation has emerged as a critical capability for deploying models in dynamic environments where training and test distributions may differ significantly. Meta-learning approaches (Finn et al., 2017; Nichol et al., 2018) learn optimization procedures that enable rapid adaptation to new tasks with limited data. More recent studies have explored lightweight methods. L-TTA (Shin & Kim, 2024) focuses on adapting only the stem layer using uncertainty minimization, while training-free methods such as TDA (Karmanov et al., 2024) leverage key-value caches to enable progressive adaptation without backpropagation. Our TMoW framework brings test-time adaptation to LM-based embodied agents by supporting both dynamic adaptation and continual expansion of world model mixtures.

## 3    APPROACH

### 3.1    OVERALL FRAMEWORK

We present the TMoW framework , which enables agents to dynamically adapt world model mixtures and continuously expand their knowledge in response to evolving environments. Similar to unified world model approaches (Zhu et al., 2025; Janner et al., 2021), each world model in the framework captures environment dynamics through a state transition function and policy.

As illustrated in Figure 2, TMoW integrates three core procedures into an MoE-based world model structure. These procedures not only allow test-time reconfiguration of world model mixtures by adjusting the mixture strategy itself, but also support efficient expansion through few-shot world model construction. (i) **Multi-granular prototype-based routing:** At test time, the most relevant

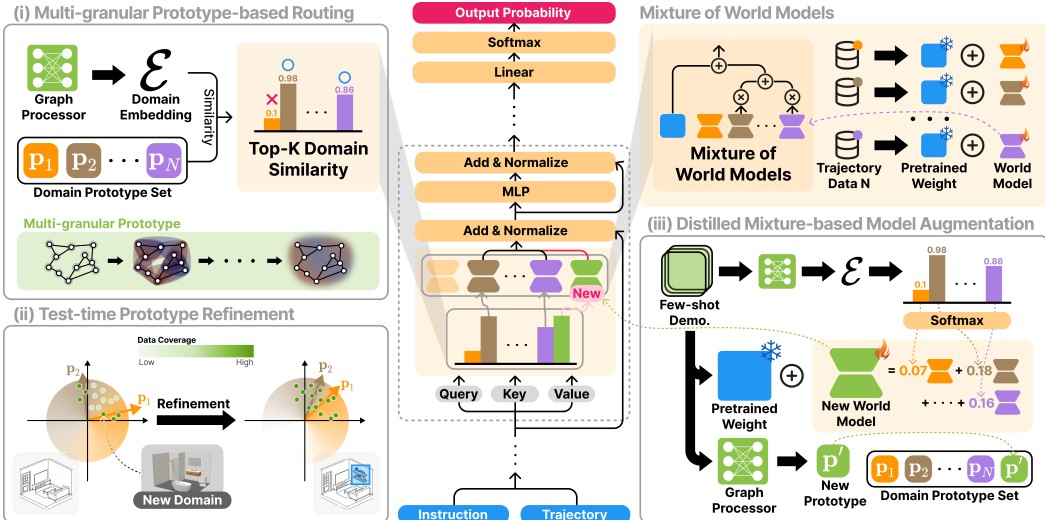

Figure 2: Overall framework of TMoW.

world models are dynamically selected and composed based on multi-granular prototypes that capture domain semantics at varying levels of abstraction, from local object interactions to global scene structures. A hierarchical message-passing network aggregates these multi-level features, enabling the layer-wise world model mixture where each granularity level can draw knowledge from different domain experts. This network works as a router across world models, allowing the framework to leverage partial domain similarities, such as shared object knowledge across different domains. (ii) **Test-time prototype refinement:** When encountering unseen domains, the router adapts by refining prototypes through weighted interpolation between existing prototypes based on similarity to the current environment. Specifically, through similarity-based refinement during ongoing environment interactions, the router expands existing prototypes to absorb new domain traits, while preserving core knowledge, thereby enabling efficient adaptation to unseen domains via dynamic reconfiguration of world model mixtures. (iii) **Distilled mixture-based model augmentation:** When encountering environments that differ significantly from seen domains, a new world model can be constructed by distilling knowledge from the mixture of existing world models based on few-shot demonstrations. This strategy leverages the collective knowledge of existing models, enabling rapid domain expansion while consolidating fragmented knowledge across diverse domains into a coherent representation of the new domain. The newly distilled world model integrates seamlessly into the framework, with its multi-granular prototype directly incorporated by the router for future mixture decisions. As such, this augmentation supports long-term expansion by introducing new world models, whereas test-time prototype refinement enables immediate adaptation by reconfiguring the mixture weights of existing models.

## 3.2 Multi-granular Prototype-based Router

**Datasets and world model.** We assume access to a pre-trained base model $M$ and demonstrations $\{\mathcal{D}_j\}_{j=1}^N$ collected from domains $\{D_j\}_{j=1}^N$. Following parameter-efficient MoE designs (Li et al., 2024), we integrate domain-specific world models into $M$ using lightweight adapters $\{m_j\}_{j=1}^N$ (e.g., LoRA (Hu et al., 2022)), which preserve the backbone parameters while enabling selective activation of multiple world models within each layer. Each adapter $m_j$ captures the environmental dynamics and policies of domain $D_j$ by learning from its corresponding $\mathcal{D}_j$. Once trained, they are combined with the base model $M$ to form a mixture of world models, denoted as $M \oplus \{m_j\}_{j=1}^N$. Each demonstration $\mathcal{D}_j = \{(i_n, \vec{\tau}_n)\}_n$ contains instruction-trajectory pairs, where $i_n \in \mathcal{I}$ specifies the task, and $\vec{\tau}_n = \{(o_t, a_t, o_{t+1})\}_t$ represents the execution trajectory consisting of observations $o_t, o_{t+1} \in \mathcal{O}$ and actions $a_t \in \mathcal{A}$.

**Multi-granular prototype.** We introduce multi-granular prototypes that capture environmental semantics across multiple levels of abstraction through hierarchical graph representations. This

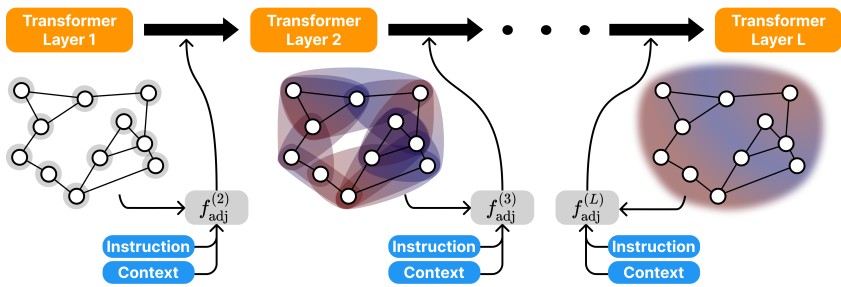

Figure 3: Graph representation in multi-granular prototype. As layers deepen, node representations evolve from local object features to global scene structures.

design exploits a key insight: while domains may differ globally, they often share common patterns at specific granularities. For instance, object interactions in kitchens and offices follow similar affordance patterns despite different houses. By maintaining representations from fine-grained node features to coarse-grained structural patterns, our multi-granular prototypes enable the layer-wise hierarchical model mixture, identifying which models share relevant patterns at each granularity.

To implement this multi-granular representation, we employ a graph processor that embeds observations into hierarchical structures, ensuring comparability at each granularity level. Given a mini-batch $\mathcal{B}$ sampled from demonstrations $\mathcal{D}_j$, the prototype at each layer $l$ for world model $j$ is computed as

$$\boldsymbol{p}_j^{(l)} = \mathop{\mathbb{E}}_{(i,\vec{\tau})\in\mathcal{B}} \mathop{\mathbb{E}}_{(o,\cdot,\cdot)\in\vec{\tau}} \left[ f^{(l)}(\mathcal{G}^{(o)}, i) \right] \tag{1}$$

where the graph processor $f^{(l)}$ transforms observation graphs $\mathcal{G}^{(o)}$ conditioned on instruction $i$. We use a Message Passing Neural Network (MPNN) (Gilmer et al., 2017) architecture, introducing a context-aware edge matrix $\tilde{\boldsymbol{E}}^{(l)}$ that dynamically adjusts neighbor aggregation based on task requirements.

$$\tilde{\boldsymbol{E}}^{(l)} = (\boldsymbol{A} + \boldsymbol{I}) \odot \boldsymbol{R} \odot f_{\text{adj}}^{(l)}(\boldsymbol{H}^{(l-1)}, i) \tag{2}$$

The adjustment function captures observation-context interactions through

$$f_{\text{adj}}^{(l)} = f_{\text{gate}}\left( (\boldsymbol{Q}_H \boldsymbol{Q}_i^T)(\boldsymbol{K}_H \boldsymbol{K}_i^T)^T / \sqrt{d} \right) \tag{3}$$

where queries $\boldsymbol{Q}_H, \boldsymbol{Q}_i$ and keys $\boldsymbol{K}_H, \boldsymbol{K}_i$ are obtained through separate layer-wise learned projections of $\boldsymbol{H}^{(l-1)}$ and the embeddings of the instruction $\Phi(i)$ respectively, and $d$ is the dimension of queries and keys.

The prototype initially contains only local information but progressively gathers neighbor information through layers, with instruction-based adjustment factors controlling this aggregation (Figure 3). This creates a natural progression from node-level features in early layers to graph-level patterns in deeper layers, analogous to how LMs progress from token-level to paragraph-level reasoning.

**Prototype-based router.** To select which world models to activate and with what routing scores, the router compares the embedding extracted from the current input data with the prototype of each world model. The router receives an instruction $i$, and observation $o$ as input and returns expert routing scores $(w_1^{(l)}, \cdots, w_N^{(l)})$ for each layer $f^{(l)}$ of the base model. Specifically, given input $(i, o)$, the routing score for expert $j$ at $l^{\text{th}}$ layer is computed as

$$w_j^{(l)} = \text{sim}(\mathcal{E}^{(l)}, \boldsymbol{p}_j^{(l)}), \ \mathcal{E}^{(l)} = f^{(l)}(\mathcal{G}^{(o)}, i) \tag{4}$$

where $\mathcal{E}^{(l)}$ is a *domain embedding* produced by the $l^{\text{th}}$ layer of the graph processor's MPNN, $\boldsymbol{p}_j^{(l)}$ is the prototype of world model $j$ at layer $l$, and $\text{sim}(\cdot, \cdot)$ denotes cosine similarity. The score vector $(\bar{w}_1^{(l)}, \cdots, \bar{w}_N^{(l)})$ at layer $l$ is sparsified by retaining only the top-$K$ entries and then normalized with a softmax and scaling hyperparameter $\tau$ ($\tau > 0$).

$$(\bar{w}_1^{(l)}, \cdots, \bar{w}_N^{(l)}) = \text{softmax}(\text{top}_K((w_1^{(l)}, \cdots, w_N^{(l)})/\tau)) \tag{5}$$

Finally, the output at $l^{\text{th}}$ layer is calculated by combining the base model with the $N$ adapters, weighted by the normalized scores $(\bar{w}_1^{(l)}, \cdots, \bar{w}_N^{(l)})$.

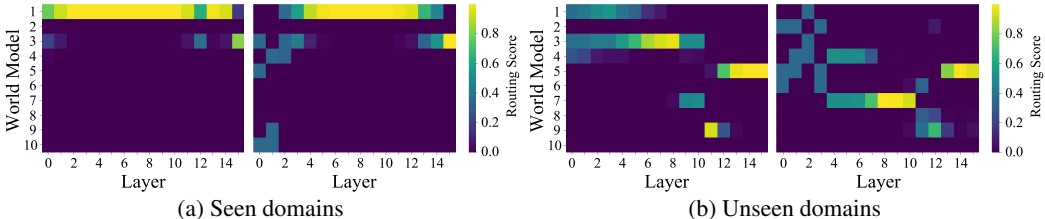

(a) Seen domains                    (b) Unseen domains

Figure 4: Comparison of the routing score heatmap before (left) and after (right) test-time prototype refinement for (a) seen and (b) unseen domains.

### 3.3 TEST-TIME PROTOTYPE REFINEMENT

We introduce test-time prototype refinement that dynamically adjusts prototype representations through environment interactions, enabling adaptation to unseen domains without retraining. Our refinement expands prototype space and densifies prototype allocation around frequently encountered features, exploiting previously underutilized knowledge within existing world models (Wang et al., 2023).

Specifically, we obtain the refined prototype $\bar{\boldsymbol{p}}_j^{(l)}$ from $\boldsymbol{p}_j^{(l)}$ using the environment interaction by

$$\bar{\boldsymbol{p}}_j^{(l)} = (1 - \alpha \operatorname{sim}(\mathcal{E}^{(l)}, \boldsymbol{p}_j^{(l)}))\boldsymbol{p}_j^{(l)} + \alpha \operatorname{sim}(\mathcal{E}^{(l)}, \boldsymbol{p}_j^{(l)})\Delta\boldsymbol{p}_j^{(l)}; \quad \Delta\boldsymbol{p}_j^{(l)} = \sum_{k=1}^{N} \bar{r}_{j,k}^{(l)}\boldsymbol{p}_k^{(l)} \quad (6)$$

where $\Delta\boldsymbol{p}_j^{(l)}$ is a refinement term, $\mathcal{E}^{(l)}$ is a domain embedding for the unseen domain from the environment interaction, and $\alpha$ is a refinement rate. The refinement term is then obtained by computing refinement weights $\bar{r}_{j,k}^{(l)}$ from prototype–prototype similarity, which are used to form the weighted sum where the weights $(\bar{r}_{j,1}^{(l)}, \cdots, \bar{r}_{j,N}^{(l)})$ are obtained through similarity measurement and softmax normalization with a scaling hyperparameter $\tau_r$ ($\tau_r > 0$):

$$(\bar{r}_{j,1}^{(l)}, \cdots, \bar{r}_{j,N}^{(l)}) = \operatorname{softmax}((r_{j,1}^{(l)}, \cdots, r_{j,N}^{(l)})/\tau_r); \quad r_{j,k}^{(l)} = \operatorname{sim}(\boldsymbol{p}_j^{(l)}, \boldsymbol{p}_k^{(l)}). \quad (7)$$

Figure 4 illustrates the comparison of routing scores before and after refinement. In the seen domain (Figure 4a), the heatmap showed minimal changes, though there is a slight tendency toward utilizing more diverse world models. In the unseen domain (Figure 4b), this diversification is more pronounced, facilitating the utilization of relatively diverse world models. The iterative prototype refinement enhances shared knowledge across world models by densifying prototype allocation around frequently encountered domains. This expands the effective coverage of the prototype space while constructing dense clusters in high-frequency regions, thereby enabling fine-grained routing strategies that foster knowledge sharing across world models.

### 3.4 DISTILLED MIXTURE-BASED MODEL AUGMENTATION

We introduce a distillation-based model augmentation that enables rapid expansion to novel domains by constructing new world models from mixtures of existing ones using few-shot demonstrations. When encountering novel domains with fundamentally different characteristics, this approach constructs new world models by distilling knowledge from weighted mixtures of existing models, where the router's weights indicate which knowledge fragments are most relevant to the unseen domain. Unlike test-time refinement, which adapts unseen domains solely by reconfiguring mixtures of existing models, this augmentation further extends the framework's capacity by generating new models that integrate seamlessly into the routing system. This seamless integration is enabled by prototype-based routing, which allows newly created models to align with existing ones and immediately join the routing process without structural changes.

Specifically, we leverage the router's weighted mixture, where the weights indicate which fragments of knowledge from models are most relevant to the unseen domain's features. Given few-shot demonstrations $\mathcal{D}' = \{(i', \vec{\tau}')\}$, we construct a combined graph $\mathcal{G}'$ from the trajectory and process it through an $f$ to obtain the layer-wise routing scores $(\bar{w}_1^{(l)}, \cdots, \bar{w}_N^{(l)})$ for each layer $l$. We initialize a new world model $m'$ as mixture of existing models, then fine-tune it on $\mathcal{D}'$ before integration into

Table 1: Zero-shot adaptation performance in VirtualHome, ALFWorld, and RLBench. Throughout the following experiments, we report 95% confidence intervals computed across 5 random seeds.

| Seen domains | VirtualHome | | ALFWorld | | RLBench | |
|---|---|---|---|---|---|---|
| Baselines | SR ($\uparrow$) | PS ($\downarrow$) | SR ($\uparrow$) | PS ($\downarrow$) | SR ($\uparrow$) | PS ($\downarrow$) |
| ZSP (Huang et al., 2022) | $10.78\%_{\pm1.50\%}$ | $27.81_{\pm0.07}$ | $2.32\%_{\pm0.19\%}$ | $49.34_{\pm0.66}$ | $11.26\%_{\pm1.59\%}$ | $17.57_{\pm0.46}$ |
| LLM+FT | $61.37\%_{\pm6.96\%}$ | $17.98_{\pm1.71}$ | $51.78\%_{\pm1.13\%}$ | $13.22_{\pm1.37}$ | $65.53\%_{\pm2.43\%}$ | $11.04_{\pm0.19}$ |
| LLM-Planner (Song et al., 2023) | $50.98\%_{\pm6.37\%}$ | $17.85_{\pm0.51}$ | $11.67\%_{\pm0.22\%}$ | $37.19_{\pm0.25}$ | $35.21\%_{\pm1.59\%}$ | $14.77_{\pm0.08}$ |
| FLARE (Kim et al., 2025) | $54.69\%_{\pm6.91\%}$ | $19.93_{\pm0.12}$ | $21.22\%_{\pm0.20\%}$ | $34.40_{\pm1.44}$ | $53.05\%_{\pm4.78\%}$ | $14.77_{\pm0.84}$ |
| SayCanPay (Hazra et al., 2024) | $64.14\%_{\pm6.17\%}$ | $15.70_{\pm0.98}$ | $51.48\%_{\pm1.72\%}$ | $13.19_{\pm0.84}$ | $68.08\%_{\pm2.93\%}$ | $8.69_{\pm0.61}$ |
| TMoW | $\mathbf{83.61\%_{\pm1.28\%}}$ | $\mathbf{11.07_{\pm0.53}}$ | $\mathbf{72.05\%_{\pm0.62\%}}$ | $\mathbf{6.94_{\pm0.21}}$ | $\mathbf{71.89\%_{\pm2.73\%}}$ | $\mathbf{6.30_{\pm0.16}}$ |

| Unseen domains | VirtualHome | | ALFWorld | | RLBench | |
|---|---|---|---|---|---|---|
| Baselines | SR ($\uparrow$) | PS ($\downarrow$) | SR ($\uparrow$) | PS ($\downarrow$) | SR ($\uparrow$) | PS ($\downarrow$) |
| ZSP (Huang et al., 2022) | $7.32\%_{\pm0.52\%}$ | $28.22_{\pm0.04}$ | $2.08\%_{\pm0.03\%}$ | $49.68_{\pm0.32}$ | $10.42\%_{\pm2.04\%}$ | $18.73_{\pm0.31}$ |
| LLM+FT | $44.24\%_{\pm6.02\%}$ | $21.00_{\pm0.39}$ | $39.61\%_{\pm0.47\%}$ | $41.24_{\pm0.47}$ | $15.63\%_{\pm3.54\%}$ | $17.44_{\pm1.10}$ |
| LLM-Planner (Song et al., 2023) | $36.05\%_{\pm0.23\%}$ | $22.93_{\pm0.01}$ | $8.46\%_{\pm0.49\%}$ | $43.54_{\pm0.11}$ | $19.79\%_{\pm4.08\%}$ | $17.19_{\pm1.54}$ |
| FLARE (Kim et al., 2025) | $40.07\%_{\pm1.02\%}$ | $22.57_{\pm0.13}$ | $11.31\%_{\pm0.45\%}$ | $42.85_{\pm0.90}$ | $34.37\%_{\pm1.80\%}$ | $11.37_{\pm0.42}$ |
| SayCanPay (Hazra et al., 2024) | $49.53\%_{\pm0.44\%}$ | $18.55_{\pm0.01}$ | $42.04\%_{\pm1.74\%}$ | $40.64_{\pm0.43}$ | $38.54\%_{\pm1.80\%}$ | $10.76_{\pm0.49}$ |
| TMoW | $\mathbf{80.16\%_{\pm1.45\%}}$ | $\mathbf{13.20_{\pm0.82}}$ | $\mathbf{68.83\%_{\pm1.15\%}}$ | $\mathbf{37.44_{\pm3.82}}$ | $\mathbf{62.75\%_{\pm2.65\%}}$ | $\mathbf{8.95_{\pm0.39}}$ |

the mixture. Then, new world model $m'$ and its corresponding prototype $p'^{(l)}$ are computed as

$$m'^{(l)} = \sum_{j=1}^{N} \bar{w}_j^{(l)} m_j^{(l)} - \eta \nabla_{m'^{(l)}} \left[ \mathbb{E}_{(\cdot,\vec{\tau}')\in\mathcal{D}'} \mathcal{L}_{\mathrm{TF}}(M \oplus m', \vec{\tau}') \right] ; \; p'^{(l)} = \mathbb{E}_{(i',\vec{\tau}')\in\mathcal{D}'} \left[ f^{(l)}(\mathcal{G}', i') \right] \quad (8)$$

where $\eta$ is learning rate, and the $\mathcal{L}_{\mathrm{TF}}$ is teacher-forcing training loss that supervises next action and observation along the trajectory, similar to Janner et al. (2021).

# 4 EXPERIMENTS

**Environments and datasets.** We evaluate TMoW with embodied environments such as VirtualHome (Puig et al., 2018), ALFWorld (Shridhar et al., 2021), and RLBench (James et al., 2020). VirtualHome is a 3D virtual environment for simulating activities in a household, ALFWorld is an indoor task simulation environment that aligns text and embodied robotic manipulation, and RLBench is a robotic manipulation benchmark for tabletop tasks, which we adapted to support language-based actions. In VirtualHome, 78 tasks (16 seen, 62 unseen) are paired with 20 distinct scenes (10 seen, 10 unseen) to form 445 demonstrations. In ALFWorld, following the CL-ALFRED benchmark (Kim et al., 2024), 6 task categories (4 seen, 2 unseen) are paired with 4 scene categories (3 seen, 1 unseen) to form 1,304 demonstrations. In RLBench, 4 task categories (3 seen, 1 unseen) are paired with 6 scene categories (4 seen, 2 unseen) to form 163 demonstrations from seen domain. Detailed descriptions of environments and datasets are provided in Appendix A.

**Evaluation Metrics.** We adopt two evaluation metrics: Success Rate (SR) and Pending Steps (PS). SR denotes the ratio of successfully completed tasks. PS stands for the average timestep taken to complete tasks, akin to cost-effectiveness in (Hazra et al., 2024).

**Baselines.** For comparison, we use five baselines. **ZSP** (Huang et al., 2022) is a zero-shot approach of using a pre-trained model to adapt to unknown domains without additional training. **LLM+FT** is an LLM that is fine-tuned with domain-specific demonstrations to improve performance in new environments. **LLM-Planner** (Song et al., 2023) is a few-shot in-context learning method to generate and refine high-level plans for new domains. **SayCanPay** (Hazra et al., 2024) is a state-of-the-art model that integrates LLMs with a heuristic cost minimization method to generate cost-effective plans. Lastly, **FLARE** (Kim et al., 2025) is a state-of-the-art embodied task planning model that combines environmental perception with adaptive replanning to generate grounded task plans by correcting predictions to align with environment features with few examples. We use Llama-3.2-3B (AI@Meta, 2024) for ZSP, LLM-Planner, FLARE, and the Say model in SayCanPay, while using trainable Llama-3.2-1B for LLM+FT, the Pay model in SayCanPay, and TMoW.

Table 2: Few-shot expansion performance in VirtualHome.

| Unseen domains, *VirtualHome* | *1-Shot* | | *5-Shot* | | *Average* | |
|---|---|---|---|---|---|---|
| Baselines | SR (↑) | PS (↓) | SR (↑) | PS (↓) | SR (↑) | PS (↓) |
| LLM+FT | 50.46%±0.44% | 19.51±0.05 | 54.36%±7.18% | 18.55±0.04 | 52.41%±3.81% | 19.03±0.04 |
| LLM-Planner (Song et al., 2023) | 40.97%±6.02% | 22.07±0.19 | 43.61%±0.92% | 21.06±0.17 | 43.30%±3.33% | 21.45±0.15 |
| FLARE (Kim et al., 2025) | 42.17%±0.37% | 22.19±0.19 | 46.64%±7.02% | 20.67±0.12 | 42.29%±3.47% | 21.56±0.18 |
| SayCanPay (Hazra et al., 2024) | 54.98%±2.09% | 17.77±0.04 | 58.88%±10.63% | 16.92±0.22 | 56.93%±6.36% | 17.35±0.13 |
| TMoW | **81.56%±1.69%** | **13.20±0.48** | **83.61%±1.33%** | **12.04±0.64** | **82.59%±1.49%** | **12.62±0.56** |

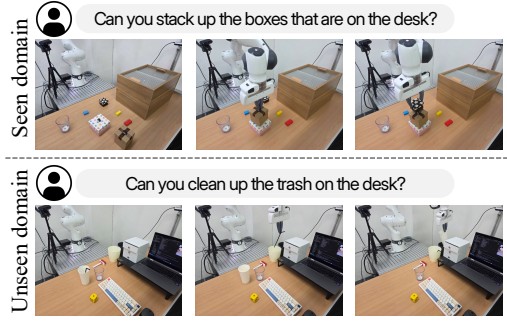

Figure 5: Examples of Real-world environment.

| Seen Domains | *Real-world* | |
|---|---|---|
| Baselines | SR (↑) | PS (↓) |
| FLARE (Kim et al., 2025) | 57.10% ± 5.99% | 7.26 ± 0.56 |
| SayCanPay (Hazra et al., 2024) | 52.90% ± 8.13% | 7.00 ± 0.37 |
| TMoW | **91.46%±2.88%** | **4.23±0.12** |
| Unseen Domains | *Real-world* | |
| Baselines | SR (↑) | PS (↓) |
| FLARE (Kim et al., 2025) | 36.04% ± 11.18% | 7.87 ± 0.56 |
| SayCanPay (Hazra et al., 2024) | 7.80% ± 4.60% | 9.71 ± 0.22 |
| TMoW | **74.64%±2.99%** | **4.89±0.18** |

Table 3: Zero-shot adaptation performance in Real-world Scenario.

## 4.1 MAIN RESULTS

**Zero-shot adaptation.** We evaluate each method in zero-shot adaptation scenarios, where agents must generalize across diverse seen and unseen domains, without access to any additional demonstrations. Each domain represents a unique combination of task and scene.

As shown in Table 1, TMoW consistently outperforms all baselines across both seen and unseen domains, particularly showing robust generalization to unseen domains. The results demonstrate the superiority of our test-time mixture approach, where multi-granular prototypes effectively capture shareable domain characteristics and enable immediate generalization through world model mixture, achieving an average improvement of 14.61% in SR and 4.42 reduction (35.31% improvement) in PS across environments for seen domains. Furthermore, test-time prototype refinement allows our model to effectively process unseen domain data by aligning unseen features with prototypes, thereby appropriately exploiting partial knowledge from existing models. This approach particularly excels in unseen domains, where we observe an average improvement of 27.21% in SR and a reduction of 3.45 steps (14.81% improvement) in PS across environments for unseen domains.

**Few-shot expansion.** We further evaluate few-shot expansion scenarios, where each target domain provides only a few demonstrations at test time. This setup examines how effectively TMoW expands its knowledge through distilled mixture-based augmentation with minimal supervision.

Table 2 compares performance across different few-shot settings (1 and 5 shots) in VirtualHome, illustrating how the number of available demonstrations affects adaptation quality. As shown, TMoW surpasses all baselines, achieving on average a 25.66% gain in SR and 4.73 step reduction in PS in VirtualHome. These results demonstrate that our distilled mixture-based augmentation efficiently achieves additional performance improvements through knowledge expansion while maintaining the modularity of the overall framework.

**Real-world scenario.** We conduct experiments in real-world environments similar to RLBench to validate the practical applicability of our approach. As shown in Figure 5, we use Franka Research 3 robot arm, and these experiments involve user instruction execution in specific object settings. In Table 3, TMoW achieves an average improvement of 34.36% in SR compared to the best baseline and 2.77 reduction (39.57% improvement) in PS for seen domains. For unseen domains, our method demonstrates even more substantial gains with an average improvement of 38.60% in SR and 2.98 reduction (37.86% improvement) in PS compared to the best performing baseline, FLARE.

Our TMoW demonstrates superior adaptation capabilities in these real-world scenarios, successfully handling tasks that require intricate manipulation sequences and environmental understanding. The results confirm that TMoW effectively transfers from simulation to reality, maintaining robust performance despite the inherent uncertainties and variations present in physical environments.

**Ablation study.** We validate TMoW's components through ablation studies in VirtualHome, Unseen domains settings. For multi-granular prototypes, we compare against TMoW-Object (using only local object features for routing) and TMoW-Scene (using only global scene features for routing) variants. To assess test-time refinement, TMoW-NoRefine uses fixed prototypes without adaptation. In Table 4a, TMoW outperforms TMoW-Object 15.49% improvement in SR and 3.60 reduction in PS, and TMoW-Scene 72.00% in SR and 14.26% in PS. This confirms that the multi-granularity approach enables efficient knowledge sharing while preserving an overall understanding of domain structure, thereby contributing to improved performance. Dynamic refinement improves SR by 7.65%, validating its effectiveness for unseen scenarios. Finally, we evaluate our distilled mixture approach against TMoW-Scratch (training from scratch). In Table 4b, our method achieves 18.39% higher SR than scratch training while using 40% less data, demonstrating superior efficiency in knowledge distillation from mixture of the existing world models.

Table 4: Ablation study of TMoW

| Model | SR ($\uparrow$) | PS ($\downarrow$) |
|---|---|---|
| *Multi-granular Prototype-based Routing* | | |
| TMoW-Object | 65.25%$\pm$4.44% | 16.72$\pm$1.99 |
| TMoW-Scene | 8.74%$\pm$0.60% | 27.38$\pm$0.33 |
| *Test-time Prototype Refinement* | | |
| TMoW-NoRefine | 73.30%$\pm$1.47% | 14.85$\pm$0.53 |
| **TMoW** | **80.74%$\pm$1.69%** | **13.12$\pm$0.87** |

(a) Zero-shot adaptation scenario

| Model | SR ($\uparrow$) | PS ($\downarrow$) |
|---|---|---|
| *Distilled Mixture-based Model Augmentation* | | |
| TMoW-Scratch | 59.84%$\pm$1.28% | 18.16$\pm$0.74 |
| **TMoW** | **81.56%$\pm$1.69%** | **13.20$\pm$0.48** |

(b) Few-shot expansion scenario

## 4.2 ANALYSIS

**Analysis for multi-granular prototype.** Figure 6a illustrates the routing score entropy across layers, revealing distinct mixture patterns of world models. We observe high entropy in early layers, indicating that multiple world models contribute equally to local object-level representations, maximizing knowledge sharing for fine-grained features. Conversely, later layers exhibit lower entropy, suggesting that global structural and scene-level embeddings become more specialized to specific world models. This hierarchical transition from shared local features to isolated global representations demonstrates how our multi-granular approach naturally balances knowledge sharing and domain specialization across different levels of abstraction.

**Analysis for test-time prototype refinement.** Figure 6b shows the routing score entropy before (TMoW-NoRefine) and after (TMoW) prototype refinement, demonstrating increased average entropy across all layers. This entropy increase reveals that refinement corrects the initial misalignment between prototypes and unseen domains, enabling the router to recognize previously underutilized knowledge within existing world models. By expanding prototype coverage to better capture unseen domain characteristics, the refinement process facilitates more active knowledge sharing across models, allowing previously underutilized models to contribute to unseen domains. This means the agent can now leverage a more diverse mixture of world model expertise.

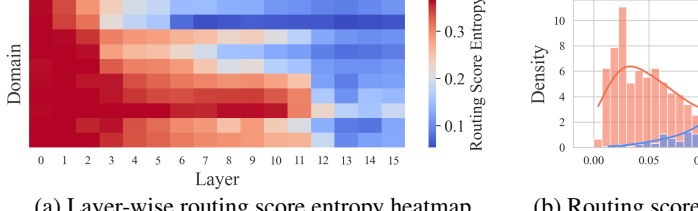

(a) Layer-wise routing score entropy heatmap    (b) Routing score entropy changes after refinement

Figure 6: Analysis for multi-granular prototype-based routing and test-time prototype refinement.

**Analysis for top-K routing.** Table 5 presents the impact of varying $K$ values in our top-$K$ routing mechanism. TMoW-Top-3 (default, same as TMoW) achieves the best overall performance (80.16% SR, 13.20 PS), significantly outperforming single-expert selection (Top-1: 65.43% SR, 17.34 PS). This validates that combining knowledge from multiple world models is essential for handling unseen domains. Notably, increasing $K$ beyond 3 leads to performance degradation, with Top-5 showing slight decline (77.52% SR, 15.19 PS) and Top-7 exhibiting substantial drops (66.01% SR, 17.67 PS), comparable to single-expert performance. This suggests that excessive expert activation introduces noise rather than useful knowledge, as irrelevant world models dilute the contribution of domain-relevant experts. The optimal $K = 3$ configuration balances knowledge diversity with focused expertise, enabling effective mixture without information overflow.

Table 5: Analysis for top-$K$ routing

| TMoW-Top-1 | | TMoW-Top-3 (TMoW) | | TMoW-Top-5 | | TMoW-Top-7 | |
|---|---|---|---|---|---|---|---|
| SR (↑) | PS (↓) | SR (↑) | PS (↓) | SR (↑) | PS (↓) | SR (↑) | PS (↓) |
| 65.43%±2.22% | 17.34±0.52 | 80.16%±1.45% | 13.20±0.82 | 77.52%±0.40% | 15.19±0.45 | 66.01%±1.33% | 17.67±0.36 |

**Continuous expansion.** Figure 7 presents our TMoW's performance in continuous domain expansion scenarios. As domains are added in new phase, the framework incorporates new world models through distilled augmentation, with refinement facilitating efficient integration. The performance improves not only for new domains but for existing ones, indicating positive knowledge transfer, while maintaining high performance on previously

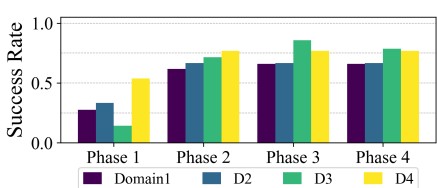

Figure 7: Continuous expansion scenario.

encountered domains without forgetting. This validates our prototype-based routing's ability to preserve existing knowledge while enabling cross-domain synergy through expanded coverage.

## 5 CONCLUSION

We presented the TMoW framework to enable embodied agents to effectively adapt to evolving environments at test time. By extending the MoE paradigm with multi-granular prototype-based routing and distilled mixture-based model augmentation, TMoW allows embodied agents to flexibly reconfigure world model mixtures at test time and efficiently construct new world models from few-shot demonstrations without retraining the entire system. Through extensive evaluation on VirtualHome and ALFWorld, we demonstrated that TMoW achieves strong performance in various adaptation scenarios, validating its effectiveness and scalability in dynamic embodied settings.

**Future work and limitation.** TMoW has two main limitations: (1) its performance is inherently bounded by the capabilities of the underlying LLM used for planning, and (2) the world model approach may face challenges in highly non-stationary environments like multi-agent settings where other agents' behaviors continuously change the environment dynamics. For future work, we plan to enhance the safety and interpretability of TMoW's routing decisions while extending it to multi-agent systems, enabling more reliable deployment and coordinated decision-making in safety-critical applications.

ACKNOWLEDGMENTS

This work was supported by the Institute of Information & Communications Technology Planning & Evaluation(IITP) grant funded by the Korea government(MSIT) (No.RS-2025-25442569, AI Star Fellowship Support Program(Sungkyunkwan Univ.), 15%, RS-2025-02218768, Accelerated Insight Reasoning via Continual Learning, 15%, RS-2022-II221045 (2022-0-01045), Self-directed multi-modal Intelligence for solving unknown, open domain problems, 15%, RS-2022-II220043 (2022-0-00043), Adaptive Personality for Intelligent Agents, 15%, RS-2020-II201821, ICT Creative Consilience Program, 15%, and RS-2019-II190421, Artificial Intelligence Graduate School Program (Sungkyunkwan University)), 15%), and IITP-ITRC (Information Technology Research Center) grant funded by the Korea government(MIST) (IITP-2024-RS-2024-00437633, 10%).

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

# A    ENVIRONMENTS

## A.1    VIRTUALHOME

We conduct our experiments using VirtualHome (Puig et al., 2018), a Unity-based simulation platform that enables embodied agents to execute complex household tasks through natural language instructions. The environment provides 20 distinct house configurations, each featuring diverse room architectures and object arrangements that capture the heterogeneity of real-world domestic spaces.

**Environment structure.**    VirtualHome represents environmental states as directed graphs, where nodes correspond to entities (agents, objects, rooms) and edges encode spatial and functional relationships. Each state observation consists of relational triples $(e_1, r, e_2)$ encoding relationships such as spatial containment (e.g., *faucet inside bathroom*), proximity (e.g., *character close plum*), adjacency (e.g., *bedroom adjacent kitchen*), and agent-object interactions (e.g., *character hold breadslice*). This graph-based representation naturally captures the compositional structure of household environments and enables reasoning about object affordances and spatial constraints.

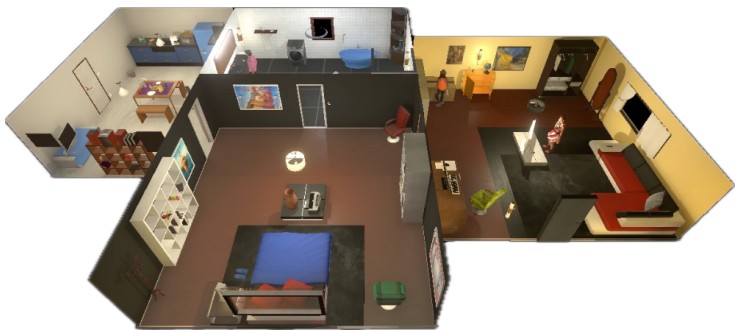

Figure A.1: The example top-view scene of the VirtualHome environment.

**Action space.**    Agents interact with the environment through six primitive action types: `walk` (navigation), `grab` (object manipulation), `open` (state changes), `put` (placement), `putin` (container interactions), and `switch` (device activation). These actions modify the environment graph according to deterministic transition rules, enabling predictable yet complex multi-step behaviors. The constrained action space reflects common household interactions while maintaining computational tractability for learning algorithms.

---

**[System]**
You are a home robot agent. You can use 6 skills, (*walk [object or room], grab [object], switch [object], open [object], putin [target object], put [target object]*). You should return only a skill after "*Action*:". Room: *livingroom, bathroom, kitchen, bedroom.*

- - - - - - - - - - - - - - - - - - - - - - - - - - - - - - - - - - - - - - - - - - - - - - - - - - - -

**[User]**
Instruction: {*instruction*}
Observation: {*observation*}
Action:

---

Figure A.2: System prompt in VirtualHome

**Task specifications.**    We evaluate on 78 household instructions spanning four task categories that test different aspects of embodied reasoning. Each instruction defines success criteria through target graph configurations, requiring agents to transform the initial environment state through appropriate action sequences. Tasks vary from simple object retrieval to complex multi-room activities involving

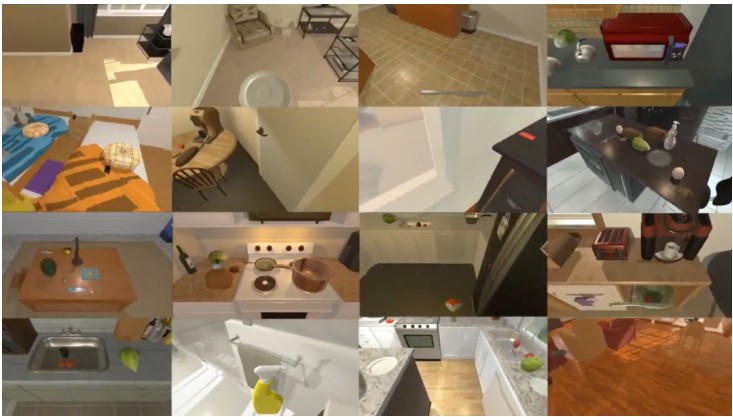

Figure A.3: The available scenes of the ALFWorld environment.

sequential dependencies and state preconditions, providing a comprehensive benchmark for evaluating adaptation capabilities across varying complexity levels.

## A.2  ALFWORLD

ALFWorld (Shridhar et al., 2021) is a text-based embodied reasoning environment that translates the ALFRED benchmark's visual tasks into natural language interactions. This environment challenges agents to execute multi-step household tasks through textual observations and instructions, emphasizing task planning and state reasoning in partially observable settings.

**Environment structure.**  ALFWorld presents environmental states through natural language descriptions that include spatial information, object locations, and agent positioning. After initialization, agents receive textual observations (e.g., *"You are in the middle of a room. Looking quickly around you, you see a armchair 1, a coffeetable 1..."*) alongside the instruction (e.g., *"Your task is to: put a pillow in armchair"*).

---

**[System]**
You are a home robot agent. You can use 10 skills, (*go to [object], take [object] from [object], put [object] on [object], open [object], close [object], toggle [object], heat [object] with [object], cool [object] with [object], clean [object] with [object], look*). You should return only a skill after "*Action*:". Room: *livingroom, bathroom, kitchen, bedroom.*

- - - - - - - - - - - - - - - - - - - - - - - - - - - - - - - - - - - - - - - - - - - - - - - -

**[User]**
Instruction: {*instruction*}
Observation: {*observation$_0$*}
Action: {*action$_0$*}
Observation: {*observation$_1$*}
Action: {*action$_1$*}
...
Action:

---

Figure A.4: System prompt in ALFWorld

**Action space.**  Agents interact with the environment through structured text commands. The action space contains navigation (`go to [location]`), perceptual query (`look`, `examine [object]`), object manipulation (`take [object]`, `put [object] in/on [receptacle]`), and state modification (`open/close [container]`, `use`

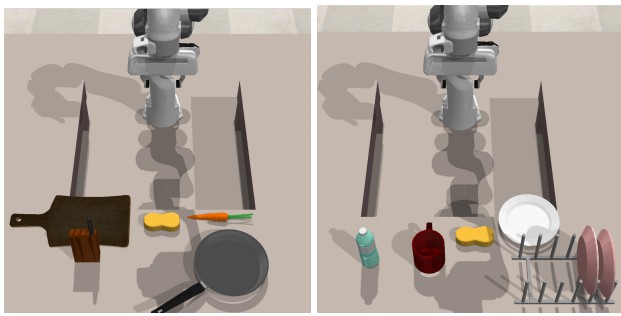

Figure A.5: The example scenes of RLBench simulator.

[appliance]). Each action triggers deterministic state transitions with corresponding textual feedback, enabling agents to track environmental changes and plan subsequent steps.

**Task specification and dataset construction.** This environment has 6 fundamental task templates that, when instantiated across various object-receptacle-room combinations, yield 1,304 unique scenarios. These tasks cover a lot of scenarios from simple object relocation to complex multi-stage procedures involving tool use. Each task requires agents to infer implicit subgoals, manage partial observability, and execute precise action sequences to achieve specified goal states. We utilize 6 task categories (4 seen, 2 unseen) with 4 scene categories (3 seen, 1 unseen) to form 1,304 episodes, and construct episodic dataset which contains task-trajectory pairs.

## A.3 RLBENCH

We evaluate in the RLBench simulator (James et al., 2020) for manipulation scenarios. Built on the CoppeliaSim simulator, RLBench provides diverse robotic platforms and a rich set of manipulation objects with varying complexity. We adapt this framework to support language-based task specification, enabling natural language instructions as input for our evaluation benchmark.

**Environment structure.** RLBench includes tables, shelves, and storage units. Across the scenes we vary furniture layouts, object types, and initial states to reflect the heterogeneity of the real environment. Each scene contains 4-6 everyday objects distributed across tables, shelves, and storage units.

**Action space.** The robot interacts through 6 primitive actions: `open [object]`, `pick [object]`, `place [object] on [object]`, `close [object]`, `wipe [object]`, `pour [object]`. Each action is executed with collision checking and inverse kinematics validation in simulation.

> **[System]**
> You are a home robot agent. You can use 6 skills, (*open [object], pick [object], place [object] on [object], close [object], wipe [object], pour [object]*). If the question is "Action:" you should answer with a skill. If the question is "Next observation:" you should answer with the next observation. You must answer only with what is requested and nothing else.
> - - - - - - - - - - - - - - - - - - - - - - - - - - - - - - - - - - - - - - - - - - - - - - - -
> **[User]**
> Instruction: {*instruction*}
> Observation: {*observation*}
> Action:

Figure A.6: System prompt in RLBench

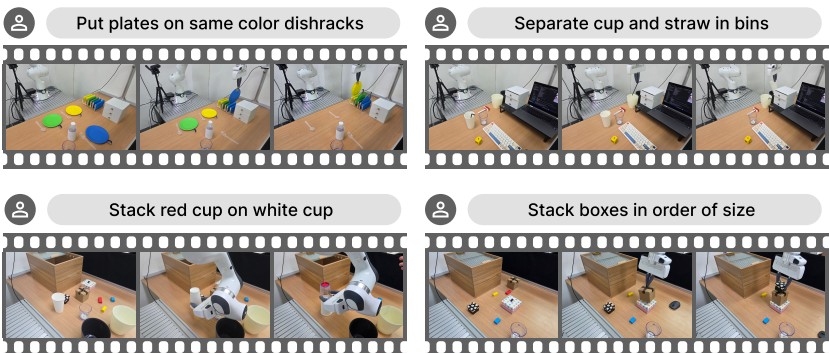

Figure A.7: The examples of the Real-world scenarios.

**Task specification and dataset construction.** Tasks span home manipulations, object relocation/organization, surface cleaning, and container handling, and specify success as satisfying a target scene graph (e.g., (yellow plate, is, clean), (yellow plate, on, yellow dishrack)). Tasks embed state preconditions and sequential dependencies (e.g., pick sponge before wipe), encouraging planning that couples perception with action. We construct datasets across 4 task categories: Place (e.g., place knife on chopboard), Pour (e.g., pour water in cup), Clean (e.g., Clean the plate) (3 seen) and Putin (e.g., Put carrot in the frying pan) (1 unseen), paired with 6 scene categories (4 seen, 2 unseen), resulting in a total of 163 episodes for the seen datasets.

## A.4 REAL-WORLD ENVIRONMENT

We evaluate in a real-world setup using a Franka Research 3 robot arm for household manipulation scenarios. Our real-world experiments assess robustness, generalization, and closed-loop adaptability under sensing uncertainty and imperfect actuation.

**Environment structure.** The workspace includes tables, shelves, and storage units. Across scenes we vary furniture layouts, object types, and initial states to reflect the heterogeneity of real homes. At episode start, the robot waits at a home pose, then uses RGB-D perception combined with VLM to extract scene-graph inference (object and relation candidates) to describe targets and constraints (e.g., container relations, surface placement, open/close capability).

**Action space.** The robot interacts through nine primitive actions: `open [object]`, `pick [object]`, `place on [object]`, `close [object]`, `wipe [object]`, `pour [object]`, `sweep [object]`, `flip [object]`, `push [object]`. Each action is executed with collision checking and inverse kinematics validation.

**Task specification and Dataset construction.** Tasks span canonical home manipulations, object relocation/organization, surface cleaning, and container handling, and specify success as satisfying a target scene graph (e.g., (yellow plate, is, clean), (yellow plate, on, yellow dishrack)). Tasks embed state preconditions and sequential dependencies (e.g., pick sponge before wipe), encouraging planning that couples perception with action. We construct 8 scenes (4 seen, 4 unseen) reflecting heterogeneous household setups, and define 3–5 tasks per scene for a total of 29 episodes. Each scene contains 8–12 everyday objects distributed across tables, shelves, and storage units.

## B IMPLEMENTATION DETAILS

In this section, we provide the implementation details of our proposed framework TMoW and each baseline. Our framework is implemented using Python v3.12 and trained on a system of an Intel(R) Core (TM) i9-10980XE processor and two NVIDIA RTX A6000 GPUs.

**[System]**
You are a home robot agent. You can use 9 skills, (*open [object], pick [object], place on [object], close [object], wipe [object], pour [object], sweep [object], flip [object], push [object]*). If the question is "Action:" you should answer with a skill. If the question is "Next observation:" you should answer with the next observation. You must answer only with what is requested and nothing else.

- - - - - - - - - - - - - - - - - - - - - - - - - - - - - - - - - - - - - - - - - - -

**[User]**
Instruction: {*instruction*}
Observation: {*observation*}
Action:

Figure A.8: System prompt in Real-world Environment

## B.1 BASELINES

**ZSP.** We use the zero-shot policy (ZSP) as a non-adaptation reference to assess the improvements achieved by TMoW. A single pretrained LLM(Llama-3.2-3B-Instruct) receives the observation from the environment, which is injected into the prompt as described in Figure A.2 and A.4. The model generates the next action step by step without any fine-tuning or additional supervision. Our implementation follows the open-source [1] with only minimal I/O adjustments to match our environment API.

**LLM+FT.** This baseline represents adaptation via fine-tuning on limited domain-specific data. It allows us to compare the efficiency and effectiveness of TMoW against conventional parameter adaptation. For the few-shot adaptation scenario, we further train the already fine-tuned model with the few-shot data from the target domain.

**LLM-Planner.** We evaluate an embodied planner that leverages in-context learning for high-level reasoning. We aim to demonstrate the effectiveness of TMoW over in-context learning through this baseline. Given the current observation and instruction, relevant demonstration snippets are concatenated with the planner prompt, then the pretrained LLM proposes the next action. For data retrieval, we use a DPR-based sentence embedding model that collects relevant data from the same dataset employed in the fine-tuning of other baselines (LLM+FT, SayCanPay). For the few-shot adaptation scenario, we augment the dataset with the target domain examples. Our implementation follows the open-source [2].

**SayCanPay.** We include a Reinforcement Learning-based planner that integrates LLM reasoning with heuristic cost minimization. By comparing against this approach, we evaluate the effectiveness of TMoW. This baseline uses three models: (1) pretrained LLM(Llama-3.2-3B-Instruct) for the Say model, (2) environment-provided optimal affordances as the Can model, and a fine-tuned LLM(Llama-3.2-1B-Instruct) for the Pay model. The hyperparameters of SayCanPay are listed in Table A.1. We follow the implementation of the open-source [3].

**FLARE.** This state-of-the-art baseline extends conventional in-context learning with an environment-adaptive replanning module that revises plans based on observed scene states. We demonstrate the superiority of TMoW in cross-domain problems by comparing with the newest method. The retriever, conditioned on the observation and instruction, collects relevant dataset examples and builds the planner prompt. If the agent fails, the model substitutes targets via semantic similarity over observed objects when names mismatch. For the few-shot adaptation scenario, target-

---

[1] https://github.com/huangwl18/language-planner
[2] https://github.com/OSU-NLP-Group/LLM-Planner
[3] https://github.com/RishiHazra/saycanpay

domain samples are additionally included in the training data. We follow the implementation of the open-source [4].

Table A.1: Hyperparameter settings and configurations of baselines

| Hyperparameter | Value |
|---|---|
| Trainable model (LLM+FT, and Pay [5]) | Llama-3.2-1B |
| Reasoning model for Simulation (ZSP, LLM-Planner, FLARE and Say [6]) | Llama-3.2-3B |
| Reasoning model for Real-world (FLARE and Say [6]) | Llama-3.2-8B |
| Batch size | 4 |
| Gradient steps | 200 |
| Learning rate scheduler | cosine |
| Initial learning rate | $5 \times 10^{-5}$ |
| Learning rate (for few-shot learning) | $1 \times 10^{-6}$ |
| Temperature (both of Llama-3.2-1B and Llama-3.2-3B) | 1.0 |

### B.2 TMoW (Ours)

Our framework, Test-time Mixture of World Models (TMoW), builds upon the parameter-efficient MoE architecture (Li et al., 2024). We employ Llama-3.2-1B as our base model with LoRA for parameter-efficient fine-tuning.

**Algorithms.** Detailed procedures of TMoW are described in Algorithms 1 through 3. Algorithm 1 describes the construction of the mixture of world models, which ranges from training each world model to extracting multi-granular prototypes. The second (Algorithm 2) contains details on test-time prototype refinement, focusing on how it works while interacting with the environment. The last algorithm (Algorithm 3) provides details of distilled model augmentation, describing the actual construction and training process of the augmented world model.

**Detailed implementation of multi-granular prototype-based router.** The multi-granular prototype-based router adopts an MPNN (Gilmer et al., 2017) structure, specifically utilizing GCN models. A key consideration in our design is the *oversmoothing* phenomenon inherent to MPNNs, where node representations become indistinguishable with increasing depth. To address this, we strategically limit MPNN layers while using standard MLPs for the remaining layers. Given that our base model (Llama-3.2-1B) contains 16 layers, we implement a hybrid architecture where the $0^{\text{th}}$, $4^{\text{th}}$, $8^{\text{th}}$, and $12^{\text{th}}$ layers employ GCN to capture graph structure at multiple granularities, while all remaining layers use standard MLPs to preserve representational diversity.

Each layer of MPNN aggregates information from neighboring nodes through three functions: message(Msg), aggregate(Agg), and update(Upd). These functions recurrently compute the hidden states for the observation graph $\mathcal{G}^{(o)} = (\boldsymbol{V}, \boldsymbol{E}, \boldsymbol{R})$ and the instruction $i$. Specifically, the hidden state of the $l^{\text{th}}$ MPNN layer is computed by

$$
\begin{aligned}
\boldsymbol{H}^{(l)} &= f^{(l)}(\mathcal{G}^{(o)}, i) \\
&= \text{Upd}^{(l)}\left( \text{Agg}^{(l)}\left( \text{Msg}^{(l)}(\boldsymbol{H}^{(l-1)}), \boldsymbol{A}, \boldsymbol{R} \right) \right)
\end{aligned}
\tag{A.1}
$$

where the initial $\boldsymbol{H}^{(0)}$ is the same as $\boldsymbol{V}$. The three functions are computed by

$$
\begin{aligned}
\tilde{\boldsymbol{M}} &= \text{Msg}^{(l)}(\boldsymbol{H}^{(l-1)}) = \boldsymbol{H}^{(l-1)}\boldsymbol{W}_M^{(l)} \\
\tilde{\boldsymbol{A}} &= \text{Agg}^{(l)}(\tilde{\boldsymbol{M}}, \boldsymbol{A}, \boldsymbol{R}) = \tilde{\boldsymbol{D}}^{-\frac{1}{2}}\tilde{\boldsymbol{E}}\tilde{\boldsymbol{D}}^{-\frac{1}{2}}\tilde{\boldsymbol{M}} \\
\boldsymbol{H}^{(l)} &= \text{Upd}^{(l)}(\tilde{\boldsymbol{A}}) = \sigma(\tilde{\boldsymbol{A}}\boldsymbol{W}_U^{(l)})
\end{aligned}
\tag{A.2}
$$

where $\tilde{\boldsymbol{D}}$ is a diagonal matrix such that $\tilde{\boldsymbol{D}}_{jj} = \sum_k \tilde{\boldsymbol{E}}_{jk}$, $\tilde{\boldsymbol{E}}$ is the context-aware edge matrix, and $\sigma$ is a sigmoid function. $\boldsymbol{W}_M^{(l)}$ and $\boldsymbol{W}_U^{(l)}$ are learnable weight matrices.

---

[4]https://github.com/snumprlab/flare

[5]Pay model in SayCanPay

[6]Say model in SayCanPay

---

**Algorithm 1** Mixture of World Models Construction

---

**Require:** Base model $M$, demonstrations $\{\mathcal{D}_j\}_{j=1}^N$, learning rate $\eta_1$ and $\eta_2$, gradient steps $T_1$ and $T_2$

**Ensure:** Mixture of world models $M \oplus \{m_j\}_{j=1}^N$, prototypes $\{\boldsymbol{p}_j^{(l)}\}_{j=1}^N$ for each layer $l$

1: **for** each domain $j \in \{1, ..., N\}$ **do**
2:     // Adapter training
3:     Initialize the adapter $m_j$
4:     **for** $step := 1$ **to** $T_1$ **do**
5:         Sample mini-batch $\mathcal{B}$ from demonstration $\mathcal{D}_j$
6:         Train adapter $m_j$ on mini-batch data $\mathcal{B}$:
7:             $m_j \leftarrow m_j - \eta_1 \nabla_{m_j} \left[ \mathbb{E}_{(\cdot, \vec{\tau}) \in \mathcal{B}} \mathcal{L}_{\text{TF}}(M \oplus m_j, \vec{\tau}) \right]$
8:     **end for**
9:     // Prototype extraction
10:     **for** each layer $l \in \{1, \cdots, L\}$ **do**
11:         Sample mini-batch $\mathcal{B}$ from demonstration $\mathcal{D}_j$
12:         Extract prototype using MPNN: $\boldsymbol{p}_j^{(l)} := \mathbb{E}_{(i,\vec{\tau}) \in \mathcal{B}} \mathbb{E}_{(o,\cdot,\cdot) \in \vec{\tau}} [f^{(l)}(\mathcal{G}^{(o)}, i)]$
13:     **end for**
14: **end for**
15: // Mixture of world models construction
16: **for** $step := 1$ **to** $T_2$ **do**
17:     Sample mini-batch $\mathcal{B}$ from a combination of the demonstrations $\cup_{j=1}^N \mathcal{D}_j$
18:     Train mixture of world models $M \oplus \{m_j\}_{j=1}^N$ on mini-batch data $\mathcal{B}$:
19:         $m_j \leftarrow m_j - \eta_2 \nabla_{m_j} \left[ \mathbb{E}_{(\cdot, \vec{\tau}) \in \mathcal{B}} \mathcal{L}_{\text{TF}}(M \oplus \{m_j\}_{j=1}^N, \vec{\tau}) \right] \quad \forall j \in \{1, \cdots, N\}$
20: **end for**
21: **return** Mixture of world models $M \oplus \{m_j\}_{j=1}^N$, prototypes $\{\boldsymbol{p}_j^{(l)}\}_{j=1}^N$ for each layer $l$

---

**Detailed implementation of context-aware edge matrix.** In equation A.2, we adjust the amount of the aggregation based on the instruction and context. The context-aware edge matrix $\tilde{\boldsymbol{E}}$ is calculated as

$$\tilde{\boldsymbol{E}} = (\boldsymbol{A} + \boldsymbol{I}) \odot \boldsymbol{R} \odot f_{\text{adj}}^{(l)}(\boldsymbol{H}^{(l-1)}, i) \tag{A.3}$$

where $\odot$ represents the Hadamard product and $\boldsymbol{I}$ is the identity matrix.

To incorporate the instruction into prototypes, we introduce an adjustment function $f_{\text{adj}}$ that modulates neighbor information aggregation based on these inputs:

$$f_{\text{adj}}^{(l)}(\boldsymbol{H}^{(l-1)}, i) = f_{\text{gate}}\left( (\boldsymbol{Q}_H \boldsymbol{Q}_i^T)(\boldsymbol{K}_H \boldsymbol{K}_i^T)^T / \sqrt{d} \right) \tag{A.4}$$

Here, $f_{\text{gate}}$ is a gate function (e.g., ReLU) and $d$ is the embedding dimension. The adjacency function employs a cross-attention mechanism that captures the interaction between observations and context. Specifically, we project the hidden states and context into query and key spaces by

$$\boldsymbol{Q}_H = \boldsymbol{H}^{(l-1)} \boldsymbol{W}_{\boldsymbol{Q}_H}^{(l)}, \boldsymbol{Q}_i = \Phi(i) \boldsymbol{W}_{\boldsymbol{Q}_i}^{(l)}; \quad \boldsymbol{K}_H = \boldsymbol{H}^{(l-1)} \boldsymbol{W}_{\boldsymbol{K}_H}^{(l)}, \boldsymbol{K}_i = \Phi(i) \boldsymbol{W}_{\boldsymbol{K}_i}^{(l)} \tag{A.5}$$

where $\boldsymbol{W}_{\boldsymbol{Q}_H}^{(l)}$, $\boldsymbol{W}_{\boldsymbol{Q}_i}^{(l)}$, $\boldsymbol{W}_{\boldsymbol{K}_H}^{(l)}$, and $\boldsymbol{W}_{\boldsymbol{K}_i}^{(l)}$ are learnable weight matrices, and $\Phi$ denotes embedding functions for instructions, e.g., from language models.

**Router pretraining.** We pretrain the router using contrastive learning with a compound loss function. For input data $\mathbf{X} = \{\mathcal{G}_1, \mathcal{G}_2, \ldots, \mathcal{G}_N\}$, we define an instance-level contrastive loss that learns augmentation-invariant representations:

$$\mathcal{L}_1(\mathbf{X}) = -\frac{1}{N} \sum_{n=1}^N \log \frac{\exp(\text{sim}(\Psi_1(\mathcal{G}_n), \Psi_2(\mathcal{G}_n))/\tau)}{\sum_{m=1}^N \exp(\text{sim}(\Psi_1(\mathcal{G}_m), \Psi_2(\mathcal{G}_n))/\tau)} \tag{A.6}$$

Additionally, we incorporate a domain-aware contrastive loss that encourages domain clustering:

$$\mathcal{L}_2(\mathbf{X}) = -\frac{1}{N} \sum_{n=1}^N \log \frac{\sum_{m=1}^N \mathbf{1}_{d[m]=d[n]} \exp(\text{sim}(\Psi_1(\mathcal{G}_m), \Psi_2(\mathcal{G}_n))/\tau)}{\sum_{m=1}^N \exp(\text{sim}(\Psi_1(\mathcal{G}_m), \Psi_2(\mathcal{G}_n))/\tau)} \tag{A.7}$$

---

**Algorithm 2** Test-time Prototype Refinement

---

**Require:** Test environment $\text{Env} : (\mathcal{O}, \mathcal{A}) \rightarrow \mathcal{O}$, instruction $i$
 1: Take an initial observation $o$ from the environment
 2: **repeat**
 3:     // Step 1: Prototype-based routing
 4:     **for** each layer $l \in \{1, \cdots, L\}$ **do**
 5:         Extract domain embedding: $\mathcal{E}^{(l)} = f^{(l)}(\mathcal{G}^{(o)}, i)$
 6:         Compute routing scores: $w_j^{(l)} = \text{sim}(\mathcal{E}^{(l)}, \boldsymbol{p}_j^{(l)})$ for all $j$
 7:         Sparsify and normalize: $(\bar{w}_1^{(l)}, \cdots, \bar{w}_N^{(l)}) = \text{softmax}(\text{top}_K((w_1^{(l)}, \cdots, w_N^{(l)})/\tau))$
 8:     **end for**
 9:     // Step 2: Mixture of world model execution
10:     $y^{(0)} := (i, o)$
11:     **for** each layer $l \in \{1, \cdots, L\}$ **do**
12:         $y^{(l)} := M^{(l)}\left(y^{(l-1)}\right) + \sum_{j=1}^{N} \bar{w}_j^{(l)} m_j^{(l)}\left(y^{(l-1)}\right)$
13:     **end for**
14:     Predicted action $a := y^{(L)}$
15:     Take next observation $o \leftarrow \text{Env}(o, a)$
16:     // Step 3: Test-time prototype refinement
17:     **for** each layer $l \in \{1, \cdots, L\}$ **do**
18:         **for** $j \in \{1, 2, \cdots, N\}$ **do**
19:             $r_{j,k} := \text{sim}(\boldsymbol{p}_j^{(l)}, \boldsymbol{p}_k^{(l)}) \quad \forall k \in \{1, \cdots, N\}$
20:             $(\bar{r}_{j,1}, \cdots, \bar{r}_{j,N}) := \text{softmax}((r_{j,1}, \cdots, r_{j,N})/\tau_r)$
21:             Compute refinement term: $\Delta\boldsymbol{p}_j^{(l)} := \sum_{k=1}^{N} r_{j,k}^{(l)} \boldsymbol{p}_k^{(l)}$
22:             Update prototypes: $\boldsymbol{p}_j^{(l)} \leftarrow (1 - \alpha \, \text{sim}(\mathcal{E}^{(l)}, \boldsymbol{p}_j^{(l)}))\boldsymbol{p}_j^{(l)} + \alpha \, \text{sim}(\mathcal{E}^{(l)}, \boldsymbol{p}_j^{(l)})\Delta\boldsymbol{p}_j^{(l)}$
23:         **end for**
24:     **end for**
25: **until** episode done

---

**Algorithm 3** Distilled Model Augmentation

---

**Require:** Few-shot demonstrations $\mathcal{D}'$, learning rate $\eta$, gradient steps $T$
**Ensure:** Distilled mixture of world models $M \oplus \{m_j\}_{j=1}^{N+1}$, prototypes $\{\boldsymbol{p}_j^{(l)}\}_{j=1}^{N+1}$ for each layer $l$
 1: **if** few-shot demonstrations $\mathcal{D}'$ available for unseen domain **then**
 2:     Construct combined graph and take instruction: $\mathcal{G}', i'$ from observations in $\mathcal{D}'$
 3:     **for** each layer $l \in \{1, \cdots, L\}$ **do**
 4:         Forward through MPNN to get routing scores: $(\bar{w}_1^{(l)}, \cdots, \bar{w}_N^{(l)})$
 5:         Initialize new adapter: $m'^{(l)} := \sum_{j=1}^{N} \bar{w}_j^{(l)} m_j^{(l)}$
 6:         Compute new prototype: $\boldsymbol{p}'^{(l)} := f^{(l)}(\mathcal{G}', i')$
 7:     **end for**
 8:     **for** $step := 1$ to $T$ **do**
 9:         Fine-tune $m'$ on $\mathcal{D}'$:
10:         $m' \leftarrow m' - \eta\nabla_{m'}\left[\mathbb{E}_{(\cdot,\vec{\tau})\in\mathcal{D}'}\mathcal{L}_{\text{TF}}(M \oplus m', \vec{\tau})\right]$
11:     **end for**
12:     $m_{N+1} := m'; \quad \boldsymbol{p}_{N+1}^{(l)} := \boldsymbol{p}'^{(l)} \quad \forall l \in \{1, \cdots, L\}$
13:     Add to model mixture: $\{m_j\}_{j=1}^{N+1}$ and layer-wise prototype set: $\{\boldsymbol{p}_j^{(l)}\}_{j=1}^{N+1}$
14: **end if**
15: **return** Mixture of world models $M \oplus \{m_j\}_{j=1}^{N+1}$, prototypes $\{\boldsymbol{p}_j^{(l)}\}_{j=1}^{N+1}$ for each layer $l$

---

The final training loss $\mathcal{L}_{\text{CL}}$ combines two contrastive losses:

$$\mathcal{L}_{\text{CL}}(\mathbf{X}) = \frac{1}{\lambda + 1}\mathcal{L}_1(\mathbf{X}) + \frac{\lambda}{\lambda + 1}\mathcal{L}_2(\mathbf{X}) \tag{A.8}$$

where $\Psi_1, \Psi_2$ are augmentation functions for creating different views, $\mathrm{sim}(\cdot, \cdot)$ is the similarity metric (e.g., cosine similarity), $d[m]$ denotes the domain label of graph $\mathcal{G}_m$, $\tau$ is the temperature parameter for contrastive learning, and $\lambda$ is the balancing coefficient between instance and domain objectives.

**Hyperparameters.** The hyperparameters of TMoW are listed in Table A.2.

Table A.2: Hyperparameter settings and configurations of TMoW training

| Hyperparameter | Value |
|---|---|
| *Common* | |
| Base model | Llama-3.2-1B |
| Learning rate scheduler | `cosine` |
| Warmup steps | 200 |
| Temperature | 1.0 |
| *World models* | |
| Batch size | 16 |
| Rank of LoRA | 32 |
| Gradient steps | 2000 |
| Initial learning rate | $1 \times 10^{-5}$ |
| *TMoW* | |
| Batch size | 1 |
| Gradient steps | 10000 |
| Initial learning rate | $1 \times 10^{-4}$ |
| Learning rate (for few-shot learning) | $1 \times 10^{-7}$ |

## C  ADDITIONAL ANALYSIS

### C.1  EXTENSION RESULTS

Table A.3: Few-shot expansion performance in VirtualHome and ALFWorld.

| **Unseen domains**, *VirtualHome* | *1-Shot* | | *5-Shot* | | *Average* | |
|---|---|---|---|---|---|---|
| Baselines | SR (↑) | PS (↓) | SR (↑) | PS (↓) | SR (↑) | PS (↓) |
| LLM+FT | 50.46%±0.44% | 19.51±0.05 | 54.36%±7.18% | 18.55±0.04 | 52.41%±3.81% | 19.03±0.04 |
| LLM-Planner (Song et al., 2023) | 40.97%±6.02% | 22.07±0.19 | 43.61%±0.92% | 21.06±0.17 | 43.30%±3.33% | 21.45±0.15 |
| FLARE (Kim et al., 2025) | 42.17%±0.37% | 22.19±0.19 | 46.64%±7.02% | 20.67±0.12 | 42.29%±3.47% | 21.56±0.18 |
| SayCanPay (Hazra et al., 2024) | 54.98%±2.09% | 17.77±0.04 | 58.88%±10.63% | 16.92±0.22 | 56.93%±6.36% | 17.35±0.13 |
| TMoW | **81.56%±1.69%** | **13.20±0.48** | **83.61%±1.33%** | **12.04±0.64** | **82.59%±1.49%** | **12.62±0.56** |

| **Unseen domains**, *ALFWorld* | *1-Shot* | | *5-Shot* | | *Average* | |
|---|---|---|---|---|---|---|
| Baselines | SR (↑) | PS (↓) | SR (↑) | PS (↓) | SR (↑) | PS (↓) |
| LLM+FT | 43.12%±2.00% | 40.18±0.32 | 48.86%±1.06% | 35.14±2.24 | 45.99%±1.53% | 37.66±1.28 |
| LLM-Planner (Song et al., 2023) | 8.91%±1.51% | 47.43±0.64 | 7.39%±0.22% | 46.79±0.00 | 8.18%±0.87% | 47.11±0.32 |
| FLARE (Kim et al., 2025) | 12.28%±0.23% | 43.67±0.88 | 11.46%±0.54% | 44.40±0.79 | 11.87%±0.39% | 44.04±0.84 |
| SayCanPay (Hazra et al., 2024) | 46.08%±1.33% | 39.00±0.33 | 49.34%±1.18% | 37.55±0.24 | 47.71%±1.92% | 38.28±0.29 |
| TMoW | **71.28%±0.05%** | **34.23±15.56** | **71.88%±0.01%** | **25.73±14.40** | **71.58%±0.03%** | **29.98±14.98** |

**Few-shot expansion scenario.** We evaluate few-shot expansion scenarios, where each target domain provides only a few demonstrations at test time. This setup examines how effectively TMoW expands its knowledge through distilled mixture-based augmentation with minimal supervision.

Table A.3 compares performance across different few-shot settings (1 and 5 shots) in VirtualHome and ALFWorld, illustrating how the number of available demonstrations affects adaptation quality. As shown, TMoW surpasses all baselines, achieving on average a 25.66% gain in SR and 4.73 step reduction in PS in VirtualHome, and a 23.87% gain in SR and 8.30 step reduction in PS in ALFWorld, compared to SayCanPay.

These results demonstrate that our distilled mixture-based augmentation efficiently achieves additional performance improvements through knowledge expansion while maintaining the modularity of the overall framework.

**Outdoor environment.** We conducted additional experiments in CARLA, an outdoor autonomous driving simulator, to validate TMoW's effectiveness in outdoor environments. We configured 6 town scenes (Town01-04 as seen, Town05, 06 as unseen). Table A.4 shows that TMoW achieves 47.49% SR and 12.03 PS, significantly outperforming baselines (FLARE: 18.34% SR, 27.51 PS). The outdoor environment presents different challenges from indoor settings, including varying weather conditions, diverse road structures, and multiple moving agents. TMoW's test-time adaptation effectively handles these environmental variations.

Table A.4: Performance comparison in CARLA outdoor environment

| Method | SR ($\uparrow$) | PS ($\downarrow$) |
|---|---|---|
| FLARE | 18.34% | 27.51 |
| TMoW (Ours) | **47.49%** | **12.03** |

**Vision-language model integration.** To enable direct processing of RGB visual observations, we integrated a pre-trained Vision-Language Model (VLM) as a graph generation module upstream of our pipeline. The VLM processes raw RGB images into text-based observations, extracting objects, relations, and states that are then converted into our graph structure.

Table A.5 presents the performance comparison between text-based and vision-based inputs in VirtualHome. Vision-based TMoW achieves 75.05% SR and 13.93 PS, showing a moderate decrease from text-based input (80.16% SR, 13.20 PS). This gap is attributed to noise in VLM's object detection and relation extraction compared to ground-truth text descriptions. However, the results demonstrate that TMoW effectively operates with real visual observations, maintaining substantial performance advantages over baselines even with noisy visual inputs.

Table A.5: Performance with Vision-Language Model integration in VirtualHome

| Input Type | SR ($\uparrow$) | PS ($\downarrow$) |
|---|---|---|
| Text-based (Ground Truth) | 80.16% | 13.20 |
| Vision-based (VLM) | 75.05% | 13.93 |

These results validate that our graph-based approach is robust to different input modalities and can be seamlessly integrated with vision systems for real-world deployment, despite the inherent noise in visual perception systems.

## C.2 MULTI-GRANULAR ROUTING FOR OVERLAPPING DOMAIN FEATURES

Our multi-granular prototype-based routing addresses domains with overlapping features at specific granularity levels. When domains share characteristics at one granularity while differing at others (e.g., office and classroom with distinct objects but similar spatial layouts), single-granularity approaches cannot leverage these complementary distinctions. Our layer-wise approach exploits these differences across abstraction levels. Early layers distinguish domains through unique local features, while deeper layers may show convergence when global scene structures are similar.

Figure A.9 demonstrates routing score distributions for domains with shared global features but distinct local objects. Initial layers show clear separation as different object types (office equipment vs. educational materials) activate distinct experts. As layers progress, routing patterns converge when domains share similar global layouts (rectangular rooms with central tables and peripheral storage). This pattern validates that our multi-granular approach captures both distinctive and shared characteristics at appropriate levels.

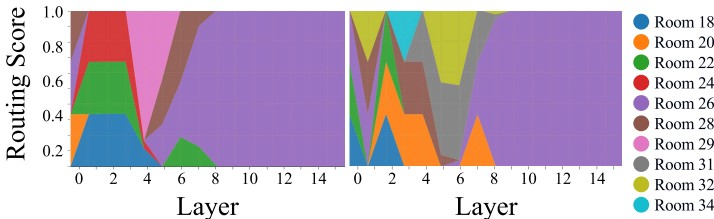

Figure A.9: The comparison of the routing score distribution.

## C.3 CORRELATION WITH ROUTING SCORE ENTROPY AND PERFORMANCE

Our analysis uncovers an insight that the routing score entropy distributions directly impact model performance. We observe that successful episodes demonstrate higher routing entropy than failed ones.

Figure A.10 compares the distribution of the routing score entropy between Success and Fail cases on unseen domains. Success cases exhibit higher entropy values (mean = 0.23, blue dashed line) compared to failed cases (mean = 0.20, red dashed line), demonstrating that distributed routing patterns correlate with task success. This indicates that leveraging diverse world models through higher entropy routing enables the framework to capture multiple domain characteristics simultaneously, leading to more robust adaptation and improved performance in complex, unseen domains.

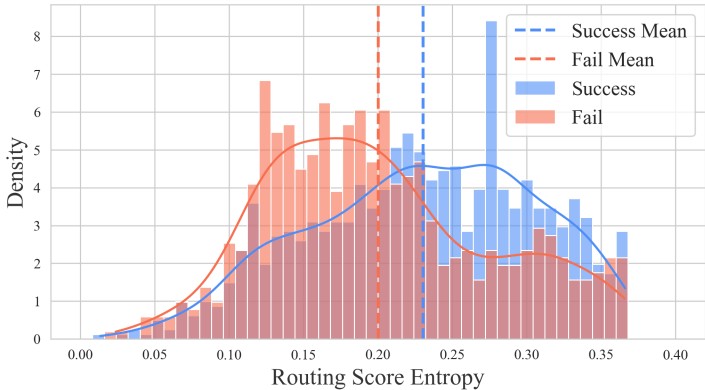

Figure A.10: The comparison of the routing score entropy distribution between success and failed case.

## C.4 ANALYSIS FOR REFINEMENT RATE $\alpha$

Figure A.11 illustrates the impact of the refinement rate $\alpha$ on success rate. When $\alpha$ is too small, insufficient refinement occurs, leading to degraded performance compared to the baseline. This suggests that with low $\alpha$ values, refinement acts as noise rather than meaningful adaptation. Due to the test-time refinement nature where updates occur during inference, insufficient learning rates require too many steps to converge, harming performance.

However, once $\alpha$ reaches a sufficient threshold ($\alpha \geq 0.5$), the model consistently achieves performance above the baseline (red dashed line) and can rapidly adapt within fewer steps. This demonstrates that while an appropriately sized refinement rate is crucial for enabling efficient test-time adaptation, our framework robustly improves performance once this threshold is met, validating the effectiveness of our approach across a range of hyperparameter settings.

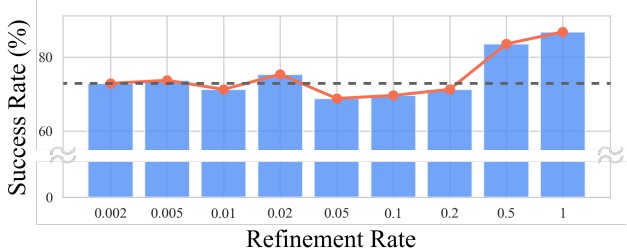

Figure A.11: The success rate per refinement rate $\alpha$ in test-time prototype refinement.

## C.5 COMPUTATION OVERHEAD

As shown in Table A.6, ZSP and LLM+FT exhibit the lowest latency, followed by our TMoW approach which achieves the next best performance. While our method uses Llama-3.2-1B in experiments, the baselines such as LLM-Planner, FLARE, SayCanPay, employ larger Llama-3.2-3B models for reasoning without training. Moreover, in-context learning methods like LLM-Planner and FLARE suffer from increased latency due to lengthy prompt processing. SayCanPay shows significantly higher latency as it requires inference across multiple models.

In contrast, our TMoW leverages adapter-based MoE architecture, enabling efficient test-time adaptation while maintaining competitive inference speed through lightweight parameter updates and selective world model routing. The prototype refinement component introduces an average overhead of only 43.40 ms, indicating that the refinement stage remains computationally minor relative to the overall inference process.

Table A.6: Average latency comparison across baselines.

| Baselines | Latency (ms) |
|---|---|
| ZSP | $115.43 \pm 24.91$ |
| LLM+FT | $115.87 \pm 24.88$ |
| LLM-Planner | $740.46 \pm 40.42$ |
| FLARE | $917.44 \pm 49.16$ |
| SayCanPay | $2470.30 \pm 106.06$ |
| TMoW-NoRefine | $656.72 \pm 15.71$ |
| TMoW | $700.12 \pm 88.41$ |

