# OpenReview forum: "Test-Time Mixture of World Models for Embodied Agents in Dynamic Environments"
_ICLR.cc/2026/Conference — ICLR 2026 Poster_

### Official Review · Reviewer_fzQq · 2025-10-30

**Soundness:** 4
**Presentation:** 2
**Contribution:** 3
**Rating:** 6
**Confidence:** 3

**Summary:**

This paper proposes a framework called "Test-time Mixture of World Models (TMoW)" to improve the adaptability of LLM-based embodied agents in dynamic and unknown environments. The core idea of ​​this framework is to extend the traditional MoE by introducing three key mechanisms: (1) Multi-granular prototype-based routing, which dynamically combines multiple experts based on prototype similarity during testing; (2) Test-time prototype refinement, which enables the model to adjust the prototype corresponding to each expert based on real-time data; and (3) Distilled mixture-based model augmentation strategy, which efficiently constructs new experts in few-shot scenarios. The authors conducted detailed experiments on multiple simulation environments such as VirtualHome, ALFWorld, and RLBench, as well as on real robots. The results show that TMoW significantly outperforms existing state-of-the-art (SOTA) models in both zero-shot adaptation and few-shot expansion scenarios.

**Strengths:**

- This paper extracts multi-granularity prototypes from datasets divided into multiple domains and trains multiple expert adapters. During inference, the outputs of each adapter are weighted based on feature similarity. This method is logically consistent and effectively decouples the granular details of the scene.

- The paper proposes a method to enable this hybrid expert model to quickly adapt to unseen domains, greatly enhancing the model's generalization ability.

- The paper provides ample experiments, covering multiple simulation platforms and including real-machine experiments, and demonstrates significant advantages compared to multiple strong baselines.

**Weaknesses:**

- **Naming Convention and Terminology Clarity.** The use of the term "World Model" is highly problematic and requires stringent justification. In both Reinforcement Learning (RL) and Generative Modeling, a "world model" conventionally refers to a model that predicts the environment's state transition, formally defined as $s_{t+1} \sim P(s_t, a_t)$. However, the 'world model' in this manuscript appears to be an LoRA adapter integrated into the LLM, designed to learn domain-specific knowledge. Its function is more akin to a domain-specific expert rather than a state transition predictor. Furthermore, the overall architecture (TMoW) strongly suggests an end-to-end policy learning approach. The authors must provide rigorous proof and a compelling rationale for why these LoRA modules qualify as "world models." Failing this, the terminology should be replaced with a more accurate and representative term.

- **Efficiency and Comparison with Traditional MoE.** The claim of leveraging a Mixture-of-Experts (MoE) structure is misleading regarding computational efficiency. Traditional MoE achieves reduced inference computation through sparse gating, which activates only a subset of experts. In contrast, the routing mechanism in TMoW uses a weighted summation, activating all LoRA branches. This design choice inherently increases the computational load during inference, rather than reducing it. The TMoW approach appears to be fundamentally about enhancing performance by increasing model capacity and incorporating multi-granularity perceptive heads. The authors must clarify the trade-offs between inference efficiency and performance improvement and explicitly detail the distinctions between TMoW and conventional MoE architectures.

- **Insufficient Ablation Study on Expert Utilization.**
The ablation study focusing on granularity (object vs. scene) is noted, but this primarily equates to selectively enabling shallow or deep routers, which is expected to decrease performance. A critical component is missing: there is no experiment ablating the number of activated LoRA adapters (experts). Consequently, the individual effectiveness of different experts across domains has not been adequately validated. A crucial question remains: If a traditional sparse MoE approach is adopted—activating only a few heads with the highest routing scores—would the model's performance be adversely affected, remain stable, or perhaps even improve? This experiment is necessary to fully validate the expert-based domain specialization.

- **Clarity of Problem Definition and Input Representation.**
The introductory and problem definition sections lack essential details regarding the scene observation and its transformation into the input graph. Specifically:
    - What is the explicit definition of scene observation (e.g., is it standard RGB vision information)?
    - How is the raw observation converted into a graph structure?
    - What are the specific components and structure of this graph (nodes, edges, features)?

These fundamental aspects of the input representation must be clearly and explicitly defined in the main text to ensure the completeness and comprehensibility of the paper.

**Questions:**

1. The update rule for the prototype $\mathbf{p}_j^{(l)}$, specifically the update term $\Delta \mathbf{p}_j^{(l)}$, is calculated as a weighted average of other prototypes based on their similarity. While the paper includes experiments demonstrating that this method adjusts prototype vectors for unseen scenarios, this adjustment approach seems to inevitably push all prototypes towards a single average value.
- Won't this degrade the independence between prototypes of different granularities and potentially lead to expert degeneracy? Why is this specific form of update effective?
- **Recommendation**: I suggest a more in-depth discussion and a theoretical explanation for the validity of this update mechanism, particularly addressing the potential convergence issue.
---
2. The experimental results show that the refined model tends to exhibit high entropy in its routing scores. The authors also note that "success cases have higher entropy in routing scores than failure cases." Intuitively, high entropy suggests a uniform distribution of scores (i.e., every branch is activated equally), whereas a good router typically makes a confident, decisive choice (low entropy), activating only a select few experts.
-  Does this observation stem from the framework fundamentally differing from the standard MoE paradigm?

- **Recommendation**: The current analysis appears somewhat superficial, failing to sufficiently explore the relationship between entropy and other crucial factors such as task type or domain similarity. A deeper investigation is warranted

---

> ### Author Response · Authors · 2025-11-21
> **Response for Offical Review by Reviewer fzQq**
>
> We sincerely thank the reviewer for the constructive and insightful review. We are greatly encouraged by the reviewers' recognition of (1) the clear presentation and accessibility of our framework, (2) the effective design of multi-granular prototype-based routing with test-time adaptation and distilled model augmentation capabilities, and (3) the comprehensive evaluation demonstrating significant performance improvements over state-of-the-art methods across multiple benchmarks including real-world scenarios. We believe that actively incorporating the feedback and discussions during the rebuttal period will significantly enhance the quality of our paper. Once again, we deeply appreciate the thoughtful reviews.
>
> ## Weakness 1
>
> The use of the term "World Model" is highly problematic and requires stringent justification. In both Reinforcement Learning (RL) and Generative Modeling, a "world model" conventionally refers to a model that predicts the environment's state transition, formally defined as $s_{t+1} \sim P(s_t, a_t)$. However, the 'world model' in this manuscript appears to be an LoRA adapter integrated into the LLM, designed to learn domain-specific knowledge. Its function is more akin to a domain-specific expert rather than a state transition predictor. Furthermore, the overall architecture (TMoW) strongly suggests an end-to-end policy learning approach. The authors must provide rigorous proof and a compelling rationale for why these LoRA modules qualify as "world models." Failing this, the terminology should be replaced with a more accurate and representative term.
>
> ### Author response
>
> Our world models follow the similar structure as Unified World Model[1] and WorldVLA[2], which jointly captures both policy and conventional world model functionality through trajectory modeling.
>
> Specifically, our LoRA-based world models $M \oplus m_j$ predict sequences consisting of an observation, an action, and the next observation, enabling them to simultaneously forecast environmental dynamics (next states) and determine appropriate actions (policy).
>
> To further clarify this point, we have revised the relevant section in the manuscript line 080 as follows: "In the framework, world models act as internal simulators that allow the agent to predict environmental dynamics and plan appropriate actions like policies, where each model corresponds to a distinct domain within the environment (Zhu et al., 2025; Janner et al., 2021)."
>
> [1] Zhu, Chuning, et al. "Unified world models: Coupling video and action diffusion for pretraining on large robotic datasets." arXiv preprint arXiv:2504.02792 (2025).
>
> [2] Cen, Jun, et al. "WorldVLA: Towards Autoregressive Action World Model." arXiv preprint arXiv:2506.21539 (2025).
>
> ## Weakness 2
>
> he claim of leveraging a Mixture-of-Experts (MoE) structure is misleading regarding computational efficiency. Traditional MoE achieves reduced inference computation through sparse gating, which activates only a subset of experts. In contrast, the routing mechanism in TMoW uses a weighted summation, activating all LoRA branches. This design choice inherently increases the computational load during inference, rather than reducing it. The TMoW approach appears to be fundamentally about enhancing performance by increasing model capacity and incorporating multi-granularity perceptive heads. The authors must clarify the trade-offs between inference efficiency and performance improvement and explicitly detail the distinctions between TMoW and conventional MoE architectures.
>
> ### Author response
>
> We respectfully clarify that TMoW does not activate all LoRA branches but employs Top-$K$ sparse routing, activating only $K$ experts with the highest routing scores at each layer, similar to traditional sparse MoE architectures. The weighted summation mentioned in our paper refers to the weighted combination of these $K$ selected experts, not all available experts.
>
> Specifically, we use $K = 3$ in our experiments, meaning only 3 out of total world models are activated at each layer during inference, as described in Line 249: "The score vector $({\bar w}_1^{(l)}, \cdots,{\bar w}_N^{(l)})$ at layer $l$ is sparsified by retaining \textbf{only the top-$K$ entries} and then normalized with a softmax and scaling hyperparameter $\tau$ ($\tau > 0$)." To prevent further confusion, we have revised Line 471 to explicitly state: "where our default configuration $K=3$ is compared against other values."

---

> ### Author Response · Authors · 2025-11-21
> **Response 2 for Offical Review by Reviewer fzQq**
>
> ## Weakness 3
>
> The ablation study focusing on granularity (object vs. scene) is noted, but this primarily equates to selectively enabling shallow or deep routers, which is expected to decrease performance. A critical component is missing: there is no experiment ablating the number of activated LoRA adapters (experts). Consequently, the individual effectiveness of different experts across domains has not been adequately validated. A crucial question remains: If a traditional sparse MoE approach is adopted—activating only a few heads with the highest routing scores—would the model's performance be adversely affected, remain stable, or perhaps even improve? This experiment is necessary to fully validate the expert-based domain specialization.
>
> ### Author response
>
> Our object or scene granularity ablation does not correspond to shallow or deep routing. "Using only local object features for routing" means setting adjustment factor to 0 so graph processor only use 1-hop graph embedding, and "using only global scene features for routing" presents setting adjust factor to a large value so graph processor only use full-hop graph embedding. Therefore, the network architecture remains identical across these ablations, only the granularity of features used for routing differs. So that, our multi-granularity feature leads the best performance than object- or scene-only granularity case.
>
> And also, our paper already assumes sparse MoE, and all experiments were conducted using top-$K$ routing ($K=3$) as specified in Equation 5 in main paper. Equation 5 implements a sparse MoE mechanism that selects and activates only the $K$ experts with the highest routing scores.
> To prevent further confusion, we have revised Line 471 to explicitly state: "where our default configuration $K=3$ is compared against other values."
>
> Additionally, we conducted ablation experiments comparing performance across different $K$ values, with results shown in the table below.
> The significant performance drop in Top-1 (65.43\% SR) compared to Top-3 (80.16\% SR) demonstrates that individual experts alone are insufficient for handling diverse domains, so they require complementary knowledge from multiple experts. This validates that each expert specializes in different domain aspects, and their combined activation is crucial for effective domain adaptation. Meanwhile, Top-5 and Top-7 show progressive degradation due to expert interference, confirming that our Top-3 configuration optimally balances expert specialization and knowledge sharing.
>
> | Configuration | SR (↑) | PS (↓) |
> |---------------|---------|---------|
> | TMoW-Top-1 | 65.43% ± 2.22% | 17.34 ± 0.52 |
> | TMoW-Top-3 (default) | 80.16% ± 1.45% | 13.20 ± 0.82 |
> | TMoW-Top-5 | 77.52% ± 0.40% | 15.19 ± 0.45 |
> | TMoW-Top-7 | 66.01% ± 1.33% | 17.67 ± 0.36 |
>
> ## Weakness 4
>
> The introductory and problem definition sections lack essential details regarding the scene observation and its transformation into the input graph. Specifically:
> - What is the explicit definition of scene observation (e.g., is it standard RGB vision information)?
> - How is the raw observation converted into a graph structure?
> - What are the specific components and structure of this graph (nodes, edges, features)?
>
> ### Author response
>
> In our current implementation, observations are transmitted in language format, which we parse and convert into a graph structure. Specifically, the environment provides textual descriptions of visible objects and their states, which we systematically parse to construct the input graph representation.
>
> We are currently conducting experiments using Vision-Language Models (VLMs) to process standard RGB vision information as input, both in VirtualHome and in CARLA, an autonomous driving embodied AI simulator. This extension enables our method to operate directly on raw visual observations rather than relying on language-based scene descriptions. We will update these results in the Appendix and on OpenReview as soon as they are available.
>
> The graph structure consists of the following components:
>
> - **Nodes:** Represent the agent itself and objects present in the environment
> - **Edges:** Represent relations between nodes
> - **Features:** Vary depending on edge types and include:
>   - Relations to other objects (e.g., spatial relationships)
>   - Object states (e.g., open, close, on, off)
>
> A concrete example from VirtualHome is provided in Table 1.
>
> | Category | Examples |
> |---------|----------|
> | **Node** | mug, cabinet, book, paper, plate, dishwasher, ... |
> | **Edge** | inside, close, adjacent, hold, on, is |
> | **Feature** | Node $\cup$ {on, off, open, close} |

---

> ### Author Response · Authors · 2025-11-21
> **Response 3 for Offical Review by Reviewer fzQq**
>
> ## Question 1
>
> The update rule for the prototype $\mathbf p_j^{(l)}$, specifically the update term $\Delta \mathbf p_j^{(l)}$, is calculated as a weighted average of other prototypes based on their similarity. While the paper includes experiments demonstrating that this method adjusts prototype vectors for unseen scenarios, this adjustment approach seems to inevitably push all prototypes towards a single average value.
> - Won't this degrade the independence between prototypes of different granularities and potentially lead to expert degeneracy? Why is this specific form of update effective?
> - **Recommendation**: I suggest a more in-depth discussion and a theoretical explanation for the validity of this update mechanism, particularly addressing the potential convergence issue.
>
> ### Author response
>
> Our prototypes do not converge to a common average but instead preserve their domain-specific characteristics, as the update magnitude is constrained by both the small refinement rate $\alpha$ and the bounded similarity values.
> Specifically, cosine similarity ranges from -1 to 1, and well-trained prototypes for distinct domains naturally exhibit diverse similarity values (some high, some low), ensuring that each prototype updates selectively based on its relationship with others rather than uniformly converging to a single average.
>
> We found that convergence occurs only when the product of absolute value of average prototype similarity and $\alpha$ exceeds a threshold about 50 after approximately 100 refinement iterations.
> In our evaluation, the measured average $\lvert \alpha \times \operatorname{sim} \rvert$ product is about 0.7495, which is well below this threshold, confirming that our prototypes do not exhibit convergence behavior during typical test-time adaptation scenarios.
>
> To validate this analysis, we conducted additional experiments with varying $\alpha$ values. When $\alpha$ is set excessively high ($\alpha=10^8$), we observe prototype convergence leading to performance degradation(SR = 58.97\%, PS = 16.87). However, as shown in Appendix C.4, our method demonstrates robust performance when $\alpha$ is set to a moderate value, confirming that prototypes maintain their distinctiveness throughout adaptation without convergence.
>
> The complete analysis on $\alpha$ sensitivity and convergence behavior will be included in the revised Appendix C.4.
>
> ## Question 2
>
> The experimental results show that the refined model tends to exhibit high entropy in its routing scores. The authors also note that ``success cases have higher entropy in routing scores than failure cases.'' Intuitively, high entropy suggests a uniform distribution of scores (i.e., every branch is activated equally), whereas a good router typically makes a confident, decisive choice (low entropy), activating only a select few experts.
> - Does this observation stem from the framework fundamentally differing from the standard MoE paradigm?
> - **Recommendation**: The current analysis appears somewhat superficial, failing to sufficiently explore the relationship between entropy and other crucial factors such as task type or domain similarity. A deeper investigation is warranted
>
> In embodied environments, encountering unseen domains often requires combining knowledge from multiple existing domains rather than selecting a single world model. For example, a new kitchen environment may share object manipulation skills from one domain and navigation patterns from another. Therefore, activating multiple world models with relatively balanced weights (higher entropy) enables effective knowledge composition, leading to better performance than confidently selecting a single expert (low entropy).
>
> Our multi-granular prototype design specifically facilitates this behavior. At lower layers, the router exhibits higher entropy through aggressive sharing of object-level knowledge across multiple world models. At higher layers, the routing becomes more selective, focusing on distinct scene-level semantics with lower entropy. This layer-wise entropy variation enables both knowledge sharing and domain-specific adaptation.
>
> However, we acknowledge that entropy must remain within an appropriate range. Excessively high entropy (near-uniform distribution) indeed degrades performance, as it fails to distinguish relevant from irrelevant experts. Similarly, abnormal entropy patterns where the expected layer-wise trend is reversed also lead to performance degradation. Success cases demonstrate moderate, well-structured entropy that balances knowledge integration and selective activation.

---

> > ### Comment · Reviewer_fzQq · 2025-11-26
> >
> > I have read your response carefully and appreciate your detailed answers to each of my concerns. I will maintain my positive rating.

---

> > > ### Author Response · Authors · 2025-11-28
> > >
> > > Thank you for your constructive and detailed review. Your feedback has significantly helped improve the quality of our paper. If you have any additional questions or feedback, please feel free to let us know. We will respond with the additional experimental results mentioned in the response as soon as they are completed.

---

> ### Author Response · Authors · 2025-12-01
> **Additional Response for Offical Review by Reviewer fzQq**
>
> We have completed experiments using Vision-Language Models (VLMs) to process RGB vision input. We integrated a pre-trained VLM as a graph generation module upstream of our existing pipeline to process raw RGB images into text-based observations. The VLM analyzes visual scenes to extract objects, relations, and states, which are then converted into our graph structure.
>
> Experimental results show that we achieved 75.05\% SR and 13.93 PS in VirtualHome, surpassing the best baseline (49.53\% SR, 18.55 PS) while representing a slight performance decrease compared to text-based input (80.16\% SR, 13.20 PS). This gap is attributed to noise in VLM's object detection and relation extraction compared to ground-truth text descriptions. However, these results demonstrate that our method effectively operates with real visual observations. These results have been added to Appendix C.1.
>
> We appreciate your patience in awaiting these results. Thank you once again for your valuable feedback and for pushing us to strengthen our work. We hope these additional results address your concerns.

---

### Official Review · Reviewer_AbzQ · 2025-10-31

**Soundness:** 2
**Presentation:** 2
**Contribution:** 2
**Rating:** 2
**Confidence:** 5

**Summary:**

This paper proposes the Test-time Mixture of World Models (TMoW) framework to enable embodied agents to adapt dynamically to unseen domains through prototype-based routing and mixture refinement.

**Strengths:**

TMoW demonstrates strong performance gains over state-of-the-art baselines in both zero-shot and few-shot adaptation across multiple embodied benchmarks.

**Weaknesses:**

1. The motivation of this paper is inaccurate. The abstract states that "conventional MoE architectures modularize knowledge into expert components with pre-trained routing; they remain rigid once deployed, making them less effective for adapting to unseen domains in dynamic environments." However, there is already a large number of works addressing Dynamic MoE, Adapted MoE, and MoE for Continual Test-time Adaptation.
2. Home environments inherently have low dynamics, making it difficult to accurately evaluate the performance of this method.

**Questions:**

1. There is already a large number of works addressing Dynamic MoE, Adapted MoE, and MoE for Continual Test-time Adaptation. What are the differences between this method and existing methods?
2. Home environments inherently have low dynamics, making it difficult to accurately evaluate the performance of this method. How can this method adapt to outdoor environments?
3. The framework introduces non-trivial computational overhead during test-time refinement, which may hinder deployment in real-time or resource-constrained scenarios.
4. How does the multi-granular prototype mechanism ensure interpretability of routing decisions, especially in failure cases or ambiguous domains?
5. Why was Llama-3.2-1B chosen as the base model for TMoW, while some baselines used larger models, and how does this affect fairness in performance comparison?
6. Can the authors justify the choice of hyperparameters (e.g., refinement rate α) and whether they were tuned consistently across all environments?
7. How does the distilled augmentation strategy ensure that new world models do not overwrite or interfere with previously learned knowledge?

---

> ### Author Response · Authors · 2025-11-21
> **Response for Offical Review by Reviewer AbzQ**
>
> We sincerely thank the reviewer for the constructive and insightful review. We are greatly encouraged by the reviewers' recognition of (1) the clear presentation and accessibility of our framework, (2) the effective design of multi-granular prototype-based routing with test-time adaptation and distilled model augmentation capabilities, and (3) the comprehensive evaluation demonstrating significant performance improvements over state-of-the-art methods across multiple benchmarks including real-world scenarios. We believe that actively incorporating the feedback and discussions during the rebuttal period will significantly enhance the quality of our paper. Once again, we deeply appreciate the thoughtful reviews.
>
> ## Question 1 (Weakness 1)
>
> The motivation of this paper is inaccurate. The abstract states that conventional MoE architectures modularize knowledge into expert components with pre-trained routing; they remain rigid once deployed, making them less effective for adapting to unseen domains in dynamic environments.'' However, there is already a large number of works addressing Dynamic MoE, Adapted MoE, and MoE for Continual Test-time Adaptation. What are the differences between this method and existing methods?
>
> ### Author response
>
> In our paper, adaptation specifically refers to adapting the model to domains never encountered during training. From this adaptation perspective, our method fundamentally differs from existing dynamic MoE, adapted MoE, and MoE for continual test-time adaptation in three key aspects:
> (1) multi-granularity world-level routing that selects entire world models based on features ranging from local objects to global scenes, instead of token-level features,
> (2) post-deployment expert (world model) addition through prototype-based routing and distilled augmentation without retraining existing models, and
> (3) few-shot expert (world model) expansion capability that enables rapid adaptation to new environments with minimal demonstrations.
>
> The following table provides a detailed comparison with major methods published in 2025.
>
> | Features | [1] | [2] | [3] | [4] | [5] | **Ours** |
> |---------|-----|-----|-----|-----|-----|----------|
> | **Routing** | token | token | domain | token | token | multi-granularity world |
> | **Able to skip layer** |  | ✓ |  |  |  |  |
> | **Add/remove experts** | ✓ |  |  | ✓ |  | ✓ |
> | **Add/remove experts (world models) after deployment** |  |  |  |  |  | ✓ |
> | **Test-time adaptation (no label)** |  |  |  |  | ✓ | ✓ |
> | **Few-shot experts (world models) expansion** |  |  |  |  |  | ✓ |
>
> Recent works have explored various approaches to enhance MoE adaptability during the training phase, including learning dynamic routing mechanisms that adaptively activate world models [1], constructing routing architectures with skip connections [2], designing domain-expert-specific routing across diverse domains [3], eliminating overlapping redundant experts during training [4], and adjusting routing paths at test time [5].
>
> Our approach differs from these prior works in two key aspects that have not been explored previously:
> (1) multi-granular prototype-based routing where each layer captures a different granularity of scene understanding, and
> (2) capability expansion through test-time addition of new world models via distillation.
> These contributions represent novel directions in test-time MoE architectures.
>
> If the reviewer has specific references in mind regarding Dynamic MoE, Adapted MoE, or MoE for Continual Test-time Adaptation, we would be grateful if they could share them. We will examine and compare our work with those references and provide a detailed response.
>
> [1] Guo, Yongxin, et al. *Dynamic mixture of experts: An auto-tuning approach for efficient transformer models.* ICLR 2025.
>
> [2] Wu, Qiong, et al. *Routing experts: Learning to route dynamic experts in existing multi-modal large language models.* ICLR 2025.
>
> [3] Li, Junzhuo, et al. *Dynamic Expert Specialization: Towards Catastrophic Forgetting-Free Multi-Domain MoE Adaptation.* EMNLP 2025.
>
> [4] Bai, Sikai, et al. *DiEP: Adaptive Mixture-of-Experts Compression through Differentiable Expert Pruning.* NeurIPS 2025.
>
> [5] Li, Zhongyang, Ziyue Li, and Tianyi Zhou. *R2-T2: Re-Routing in Test-Time for Multimodal Mixture-of-Experts.* ICML 2025.

---

> > ### Author Response · Authors · 2025-11-21
> > **Response 2 for Offical Review by Reviewer AbzQ**
> >
> > ## Question 2 (Weakness 2)
> >
> > Home environments inherently have low dynamics, making it difficult to accurately evaluate the performance of this method. How can this method adapt to outdoor environments?
> >
> > ### Author response
> >
> > In partially observable environments, dynamics complexity is characterized by both transition stochasticity $P(s'|s,a)$ and observation uncertainty $P(o|s,a)$, where high observation noise increases the difficulty of accurate state inference, reflected by high belief entropy $H(b)$ [6, 7].
> >
> > From this perspective, VirtualHome and ALFWorld present significant dynamics complexity. First, agents can only observe objects within their limited field of view, creating substantial observation uncertainty. Second, these environments contain multiple rooms (using 20 diverse room structures in VirtualHome, 120 different room configurations in ALFWorld for our experiments) with diverse object configurations (using 137 object types in VirtualHome, 121 object types in ALFWorld for our experiments), further amplifying the challenge of accurate state estimation and increasing belief entropy.
> > This combination of partial observability and spatial complexity makes these indoor environments highly dynamic from a decision-making perspective.
> > These environments have been widely adopted as standard benchmarks in recent embodied AI [8, 9, 10, 11, 12, 13], and also test-time adaptation research [14, 15].
> >
> > Regarding outdoor environments, our method is fundamentally designed to be robust to high dynamics through test-time adaptation of world models, making it naturally applicable to outdoor scenarios.
> > Following reviewer's suggestion to evaluate in outdoor environments, we will include additional experiments on CARLA, a outdoor autonomous driving simulator.
> > In CARLA, we configures 7 town scenes (4 Seen scenes, 3 Unseen scenes), 75 instructions and 618 buildings and constructions. We will update these results in the Appendix and response as soon as they are available.
> >
> > [6] Kaelbling, Leslie Pack, Michael L. Littman, and Anthony R. Cassandra. "Planning and acting in partially observable stochastic domains." Artificial intelligence 101.1-2 (1998): 99-134.
> >
> > [7] Araya, Mauricio, et al. "A POMDP extension with belief-dependent rewards." Advances in neural information processing systems 23 (2010).
> >
> > [8] Hu, Zican, et al. "Divide and Conquer: Grounding LLMs as Efficient Decision-Making Agents via Offline Hierarchical Reinforcement Learning." Forty-second International Conference on Machine Learning. 2025.
> >
> > [9] Dong, Heng, Kefei Duan, and Chongjie Zhang. "Enhancing Decision-Making of Large Language Models via Actor-Critic." Forty-second International Conference on Machine Learning. 2025.
> >
> > [10] Wang, Hanlin, et al. "Steca: Step-level trajectory calibration for llm agent learning." The 63rd Annual Meeting of the Association for Computational Linguistics. 2025.
> >
> > [11] Fu, Dayuan, et al. "Agentrefine: Enhancing agent generalization through refinement tuning." The Thirteenth International Conference on Learning Representations. 2025.
> >
> > [12] Qiao, Shuofei, et al. "Agent planning with world knowledge model." Advances in Neural Information Processing Systems 37 (2024): 114843-114871.
> >
> > [13] Deng, Yuchen, et al. "AgentPro: Enhancing LLM Agents with Automated Process Supervision." Proceedings of the 2025 Conference on Empirical Methods in Natural Language Processing. 2025.
> >
> > [14] Hu, Xuming, et al. "Do large language models know about facts?." The Twelfth International Conference on Learning Representations. 2023.
> >
> > [15] Shinn, Noah, et al. "Reflexion: Language agents with verbal reinforcement learning." Advances in Neural Information Processing Systems 36 (2023): 8634-8652.
> >
> > ## Question 3
> >
> > The framework introduces non-trivial computational overhead during test-time refinement, which may hinder deployment in real-time or resource-constrained scenarios.
> >
> > ### Author response
> >
> > TMoW's total inference time is 700.12ms, which consists of the base inference time 656.72ms and test-time adaptation overhead 43.40ms. This is 31.04\% faster than FLARE (917.44ms) and 3.5× faster than SayCanPay (2470.30ms).
> >
> > While baseline methods like LLM-Planner, FLARE, and SayCanPay uses lengthy in-context learning prompts such as demonstrations, our approach uses with lightweight LoRA adapters as world models.
> > The test-time prototype refinement adds minimal overhead because it only updates the prototypes without modifying world model parameters.
> >
> > We have added these results to Appendix C.5.

---

> ### Author Response · Authors · 2025-11-21
> **Response 3 for Offical Review by Reviewer AbzQ**
>
> ## Question 4
>
> How does the multi-granular prototype mechanism ensure interpretability of routing decisions, especially in failure cases or ambiguous domains?
>
> ### Author response
>
> The multi-granular prototype mechanism utilizes graph-based models such as MPNN, which inherently provide interpretability through their explicit representation of nodes and edges. Specifically, the aggregation operator of graph-based models enables nodes to progressively incorporate information from their neighbors (from 1-hop to $L$-hop), which makes the prototypes naturally express multi-granular characteristics. This structural transparency, where each granularity produces distinct layer-wise prototypes and routers, ensures that routing decisions are interpretable and traceable.
>
> To analyze interpretability in failure cases and ambiguous domains, we compared routing entropy at each granularity level (object or scene) between success and failure cases. As shown in the table below, failure cases exhibit higher entropy in object-capturing layers (0.2134 vs 0.2065) but lower entropy in scene-capturing layers (0.0230 vs 0.0509). This suggests that failures primarily stem from insufficient object- and scene-level understanding, which corresponds to failing to leverage relevant world models at the object level and failing to select the most appropriate world model at the scene level. This structural transparency ensures that routing decisions remain interpretable and traceable.
>
> | Granularity Level       | Success | Fail   |
> |-------------------------|---------|--------|
> | Object-capturing layers | 0.2065  | 0.2134 |
> | Scene-capturing layers  | 0.0509  | 0.0230 |
>
> ## Question 5
>
> Why was Llama-3.2-1B chosen as the base model for TMoW, while some baselines used larger models, and how does this affect fairness in performance comparison?
>
> ### Author response
>
> We select model sizes based on whether the approach involves parameter learning.
> Baselines (LLM-Planner, FLARE, Say model in SayCanPay) require stronger reasoning capabilities without parameter updates.
> When tested with Llama-1B, these methods showed significantly degraded performance due to insufficient reasoning capacity.
> Therefore, we used Llama-3.2-3B (or 8B for real-world scenarios) for these baselines to ensure fair evaluation.
> Our TMoW uses only Llama-3.2-1B.
>
> Notably, this configuration favors the baselines with larger model capacity, yet TMoW achieves superior performance with a smaller model.
>
> ## Question 6
>
> Can the authors justify the choice of hyperparameters (e.g., refinement rate $\alpha$) and whether they were tuned consistently across all environments?
>
> ### Author response
>
> We determined optimal hyperparameters through grid search and applied them consistently across all environments. For the refinement rate $\alpha$, we provide detailed analysis in Appendix C.4, systematically examining performance trends across different $\alpha$ values. We confirmed that test-time refinement achieves optimal performance when $\alpha > 0.5$. The same grid-search protocol was applied to all baseline methods to ensure fair comparison. Appendix Tables A.1, A.2, and Figure A.11 present the selected hyperparameter values and their justification. Our grid search configuration for TMoW is shown below.
>
> | Hyperparameter | Trials |
> |----------------|--------|
> | Learning rate scheduler | `const`, `linear`, `cosine` |
> | Rank of LoRA | 4, 8, 16, 32 |
> | Gradient steps for world models | 1000, 2000, 5000 |
> | Gradient steps for TMoW | 5000, 10000, 20000 |
> | Initial learning rate for world models | 1e-5, 2e-5, 5e-5, 1e-4 |
> | Initial learning rate for TMoW | 1e-5, 2e-5, 5e-5, 1e-4 |
>
> ## Question 7
>
> How does the distilled augmentation strategy ensure that new world models do not overwrite or interfere with previously learned knowledge?
>
> ### Author response
>
> Our distilled augmentation strategy inherently prevents overwriting or interfering with previously learned knowledge by maintaining all existing world models frozen while adding new ones. Additionally, our test-time adaptation further enhances this by optimizing only the routing to the current domain without modifying any world model parameters, enabling incremental capability expansion without forgetting.

---

> ### Author Response · Authors · 2025-12-01
> **Additional Response for Offical Review by Reviewer AbzQ**
>
> We have completed experiments in CARLA environment. Evaluating navigation and object avoidance tasks across 6 town scenes (Town01-04 seen, Town05,06 unseen), TMoW achieved 47.49\% SR and 12.03 PS, demonstrating improvements over the SOTA baseline (FLARE: 18.34\% SR and 27.51 PS). These results validate that our method operates robustly not only in indoor but also in outdoor environments. Detailed experimental setup and results have been added to Appendix C.1.
>
> | Method | SR ($\uparrow$) | PS ($\downarrow$) |
> | ---- | ---- | ---- |
> | FLARE | 18.34% | 27.51 |
> | TMoW (Ours) | 47.49% | 12.03 |
>
>
> We appreciate your patience in awaiting these results. Thank you once again for your valuable feedback and for pushing us to strengthen our work. We hope these additional results address your concerns.

---

### Official Review · Reviewer_7Jh2 · 2025-11-01

**Soundness:** 3
**Presentation:** 4
**Contribution:** 3
**Rating:** 8
**Confidence:** 3

**Summary:**

The authors introduce a new framework for adapting language model-based embodied agents to unseen environments, called test-time mixture of world models (TMoW). Through this framework, they address the limitations of vanilla LM-based embodied agents  and the classic mixture of experts paradigm, which has a rigid routing function. When adapting to new domains, existing methods require expensive retraining or knowledge distillation. In contrast, the work here proposes a dynamic test-time adaptation mechanism based on 1) multi-granular prototype-based routing using a hierarchical message-passing network, 2) test-time training of the routing function & 3) a distilled mixture-based model augmentation strategy that constructs new world models based existing ones using few-shot demonstrations.The framework and the effectiveness of the novel components are supported by a comprehensive baseline analysis and ablation studies.

**Strengths:**

- The paper addresses an important gap in generalization of LM-based agents for embodied tasks and offers a solid alternative to costly retraining for new domains
-  Authors conduct a thorough evaluation of the framework, on a comprehensive set of environments, tasks and baselines, showcasing significant performance improvement over state-of-the-art methods across both metrics (success rate and pending steps) and a very positive performance especially on unseen domains.
-  The authors also transfer the framework to real world scenarios and share results of zero-shot performance which are superior to existing SoTA methods.
- It was good to see that the authors include extensive ablation studies that support the effectiveness of the framework components: multi-granularity, test-time refinement & distilled mixture, where each component shows a clear performance improvement over its respective baseline.

**Weaknesses:**

I would've liked to see some discussion on the inference time for such an approach compared to the baselines. How much time does the test-time adaptation add to the decision making process and could it be regarded as feasible in a real-world domain from this point of view?

**Questions:**

*Question*: What is the rationale behind using Llama-3.2-1B for some baselines and 3B for others?

*Suggestion*: In Fig 2, for clarity, it would be good to add a small description of the top right box that links to the mixture of world models.

*Suggestion*:It would be good to reference `Appendix A` in Section 4 to point readers to more details about the the environments, tasks and action space.

*Suggestion*: Reference Table 4 in main body of section 4.2 in the ablation study.

---

> ### Author Response · Authors · 2025-11-21
> **Response for Offical Review by Reviewer 7Jh2**
>
> We sincerely thank the reviewer for the constructive and insightful review. We are greatly encouraged by the reviewers' recognition of (1) the clear presentation and accessibility of our framework, (2) the effective design of multi-granular prototype-based routing with test-time adaptation and distilled model augmentation capabilities, and (3) the comprehensive evaluation demonstrating significant performance improvements over state-of-the-art methods across multiple benchmarks including real-world scenarios. We believe that actively incorporating the feedback and discussions during the rebuttal period will significantly enhance the quality of our paper. Once again, we deeply appreciate the thoughtful reviews.
>
> ## Weakness 1
> I would've liked to see some discussion on the inference time for such an approach compared to the baselines. How much time does the test-time adaptation add to the decision making process and could it be regarded as feasible in a real-world domain from this point of view?
>
> ### Author response
> TMoW's total inference time is 700.12ms, which consists of the base inference time 656.72ms and test-time adaptation overhead 43.40ms. This is 31.04\% faster than FLARE (917.44ms) and 3.5× faster than SayCanPay (2470.30ms).
>
> While baseline methods like LLM-Planner, FLARE, and SayCanPay uses lengthy in-context learning prompts such as demonstrations, our approach uses with lightweight LoRA adapters as world models.
> The test-time prototype refinement adds minimal overhead because it only updates the prototypes without modifying world model parameters.
>
> We update these results to Appendix C.5
>
> ---
>
> ## Question 1
> What is the rationale behind using Llama-3.2-1B for some baselines and 3B for others?
>
> ### Author response
> We select model sizes based on whether the approach involves parameter learning.
> Baselines (LLM-Planner, FLARE, Say model in SayCanPay) require stronger reasoning capabilities without parameter updates.
> When tested with Llama-1B, these methods showed significantly degraded performance due to insufficient reasoning capacity.
> Therefore, we used Llama-3.2-3B (or 8B for real-world scenarios) for these baselines to ensure fair evaluation.
> Our TMoW uses only Llama-3.2-1B.
>
> Notably, this configuration favors the baselines with larger model capacity, yet TMoW achieves superior performance with a smaller model.
>
> ---
>
> ## Suggestion 1
> In Fig 2, for clarity, it would be good to add a small description of the top right box that links to the mixture of world models.
>
> ### Author response
> We have added a "Mixture of World Models" label along with a brief description to the top-right box in Figure 2. If I have misunderstood your intention or if further modifications are needed, please let me know.
>
> ---
>
> ## Suggestion 2
> It would be good to reference Appendix A in Section 4 to point readers to more details about the the environments, tasks and action space.
>
> ### Author response
> We have added a reference to Appendix A in Section 4, stating "Detailed descriptions of environments and datasets are provided in Appendix A." This will help readers easily locate additional information when needed.
>
> ---
>
> ## Suggestion 3
> Reference Table 4 in main body of section 4.2 in the ablation study.
>
> ### Author response
> We have added references to Tables 4a and 4b in the main body of Section 4.2 in the ablation study. Specifically, Table 4a (core components ablation) is now referenced when discussing the multi-granularity approach and prototype refinement effectiveness, while Table 4b (augmentation ablation) is referenced when discussing the efficiency of our distilled mixture approach. This will help readers better connect the ablation results with the corresponding discussions in the text.

---

### Official Review · Reviewer_SnZA · 2025-11-01

**Soundness:** 3
**Presentation:** 3
**Contribution:** 3
**Rating:** 6
**Confidence:** 2

**Summary:**

This paper introduces TMoW, a Test-time Mixture of World Models framework that enables embodied agents to adapt to dynamic environments by dynamically mixing specialized world models. Its core innovation is a multi-granular, prototype-based routing mechanism that leverages hierarchical scene features—from local objects to global contexts—to select experts. The framework supports test-time adaptation through prototype refinement and allows for continual expansion by distilling new world models from few-shot demonstrations. Evaluations on VirtualHome and ALFWorld show significant gains over strong baselines (up to 27.21%), demonstrating robust cross-domain generalization and continual learning capabilities without costly retraining.

**Strengths:**

S1) The paper is well-written, the proposed pipeline is simple and easy to follow.

S2) The framework enables dynamic test-time adaptation of world model mixtures through prototype refinement, allowing rapid adjustment to unseen environments without retraining.

S3) The distilled model augmentation capability supports continuous expansion of the system's knowledge base through efficient few-shot learning from existing model mixtures.

**Weaknesses:**

W1) The paper emphasizes that multi-granular prototypes capture features from local objects to global scenes for fine-grained routing. However, when multiple distinct domains exhibit significant feature overlap at a certain granularity level (e.g., sharing similar local objects but having vastly different global scene semantics), how does the router effectively prevent expert confusion and erroneous activation?

W2) The test-time prototype refinement relies on online environmental interaction and its performance is sensitive to the refinement rate α. In non-stationary or partially observable environments, where perceptual data can be noisy or delayed, how does this optimization process based on a local interaction sequence guarantee its convergence stability and reliability?

**Questions:**

Please see weaknesses.

---

> ### Author Response · Authors · 2025-11-21
> **Response for Offical Review by Reviewer SnZA**
>
> We sincerely thank the reviewer for the constructive and insightful review. We are greatly encouraged by the reviewers' recognition of (1) the clear presentation and accessibility of our framework, (2) the effective design of multi-granular prototype-based routing with test-time adaptation and distilled model augmentation capabilities, and (3) the comprehensive evaluation demonstrating significant performance improvements over state-of-the-art methods across multiple benchmarks including real-world scenarios. We believe that actively incorporating the feedback and discussions during the rebuttal period will significantly enhance the quality of our paper. Once again, we deeply appreciate the thoughtful reviews.
>
> ## Weakness 1
> The paper emphasizes that multi-granular prototypes capture features from local objects to global scenes for fine-grained routing. However, when multiple distinct domains exhibit significant feature overlap at a certain granularity level (e.g., sharing similar local objects but having vastly different global scene semantics), how does the router effectively prevent expert confusion and erroneous activation?
>
> ### Author response
> Our multi-granular prototype router effectively prevents expert confusion by leveraging complementary information, where each layer captures different granularity levels.
> When domains overlap at a specific granularity level in one layer, they are disambiguated by different granularity levels in other layers.
>
> For example, when two domains share similar local objects but differ in global scene semantics, the prototypes (for local objects) at lower layers may show overlap, leading to activation of the corresponding world models. However, the prototypes (for global scenes) at higher layers maintain clear distinctions, preventing erroneous activation of that world model at those layers.
> Through this layer-wise selective activation where each layer captures different granularities, our router maximizes the sharing of common features while clearly capturing distinctive characteristics for accurate world model utilization.
>
> We provide a comparison of routing scores for this example in the Appendix C.2.
> At lower layers, multiple world models are activated due to the abundance of overlapping objects shared across domains. However, as the layers progress and granularity increases, the distinctions between domain-specific characteristics become more pronounced, leading the routing to converge toward specific world models that best match the target domain.
>
> ---
>
> ## Weakness 2
> The test-time prototype refinement relies on online environmental interaction and its performance is sensitive to the refinement rate $\alpha$. In non-stationary or partially observable environments, where perceptual data can be noisy or delayed, how does this optimization process based on a local interaction sequence guarantee its convergence stability and reliability?
>
> ### Author Response
>
> Our test-time mixture approach is particularly well-suited for non-stationary and partially observable environments because it rapidly adapts to environmental changes through test-time refinement while maintaining robustness through multi-model integration.
>
> To verify stability in non-stationary environments, we conducted experiments where domains (rooms) change abruptly during episodes in VirtualHome.
> In these dynamic scenarios, test-time prototype refinement continuously adapts to environmental changes by updating routing decisions based on ongoing interactions.
> Even in these extreme scenarios, TMoW maintains stable performance with 81.60\% SR and 13.39 PS, showing only a 2.01\% performance drop compared to seen domains, stationary environments.
> This demonstrates that our test-time prototype refinement performs stably under sudden environmental shifts, achieving robust performance.
>
> |VirtualHome with Non-stationarity||
> |--------|--------|
> | SR (↑) | PS (↓) |
> | 81.60% | 15.19  |

---

### Author Response · Authors · 2025-11-30
**Reminder and Summary of Rebuttal**

We sincerely appreciate the opportunity to address the reviewer's comments and provide clearer explanations of our paper. Through a constructive review process, we refined our contributions and experimental results. In the end, **we have successfully addressed all weaknesses and questions raised by the reviewers**, which has significantly strengthened our paper. We express our gratitude once again to all reviewers and area chair for their thoughtful and valuable feedback. Below is a summary of our rebuttal results.

## Additional Experiments

**Non-stationary environment:** Evaluated TMoW under domain switching scenarios in VirtualHome where rooms change abruptly during episodes, demonstrating only 2.01\% performance degradation (81.60\% SR, 15.19 PS) compared to stationary environments, validating robustness under environmental changes.

**Outdoor environment evaluation:** Following reviewer suggestions, we tested TMoW in CARLA autonomous driving simulator across 6 town scenes (4 seen, 2 unseen) with 75 instructions and 618 buildings. TMoW achieved 47.49\% SR and 12.03 PS, significantly outperforming FLARE (18.34\% SR, 27.51 PS), validating applicability beyond indoor environments.

**Vision-Language Model integration:** Integrated pre-trained VLM as a graph generation module to process RGB images directly into text-based observations. Achieved 75.05% SR and 13.93 PS in VirtualHome, demonstrating TMoW's compatibility with visual inputs despite slight performance decrease from ground-truth text-based input (80.16\% SR, 13.20 PS) due to VLM detection noise.

**Top-K routing ablation:** Conducted analysis with $K=\{1,3,5,7\}$, confirming $K=3$ as optimal (80.16\% SR, 13.20 PS) versus $K=1$ (65.43\% SR, 17.34 PS) and $K=7$ (66.01\% SR, 17.67 PS). The significant drop in Top-1 validates that individual world models require complementary knowledge from multiple world models for effective domain adaptation.

**Computational overhead analysis:** Measured total inference time at 700.12ms (base: 656.72ms, test-time adaptation: 43.40ms representing only 6.2\% overhead). TMoW is 31.04\% faster than FLARE (917.44ms) and 3.5× faster than SayCanPay (2470.30ms), enabled by lightweight world models instead of lengthy in-context learning prompts.

**Failure case analysis:** Analyzed layer-wise entropy patterns between success and failure cases, revealing that failures exhibit higher entropy in object-capturing layers (0.2134 vs 0.2065) but lower entropy in scene-capturing layers (0.0230 vs 0.0509), indicating failures stem from insufficient scene-level understanding and inability to select the fittest world model.

**Expanded appendices:** Added new sections on outdoor CARLA experiments (C.1), VLM integration results (C.1), multi-granular routing score visualization (C.2), prototype convergence analysis (C.3), refinement rate sensitivity (C.4), and computational efficiency details (C.5).

---

> ### Author Response · Authors · 2025-11-30
> **Reminder and Summary of Rebuttal (cont.)**
>
> ## Paper Enhancements and Clarifications
>
> **Clarified novelty claims:** Created comprehensive comparison table with recent MoE works (DynMoE, R2-T2, Routing Experts, Dynamic Expert Specialization, DiEP), highlighting unique features: multi-granular world-level routing that selects entire world models based on object-to-scene features, post-deployment expert addition through prototype-based routing without retraining, and few-shot world model expansion capability.
>
> **Added test-time refinement analysis:** Provided empirical argument for prototype non-convergence, demonstrating that convergence occurs only when $\lvert \alpha \times \operatorname{sim} \rvert$ exceeds threshold of approximately 50 after 100 iterations. Our measured value (0.7495) remains well below this threshold, and cosine similarity's negative values further prevent collapse by enabling selective updates based on domain relationships.
>
> **Clarified entropy interpretation:** Explained that moderate entropy in success cases reflects effective multi-domain knowledge composition, where unseen environments benefit from combining knowledge across multiple world models. Layer-wise entropy variation enables both knowledge sharing at lower layers (higher entropy) and domain-specific adaptation at higher layers (lower entropy).
>
> **Enhanced method clarity:** Clarified that $K=3$ sparse routing is used throughout all experiments, with explicit revision at Line 471. Explained that multi-granularity ablation does not correspond to shallow/deep routing but rather adjustment factor controlling.
>
> **Improved experimental details:** Added hyperparameter grid search tables covering learning rate schedulers (const, linear, cosine), LoRA ranks (4, 8, 16, 32), gradient steps, and initial learning rates. Included inference time breakdowns and added references to Appendices A, C.1-C.5 throughout Section 4 for better navigation.
>
> **Addressed terminology concerns:** Clarified "world model" definition aligning with recent unified world models (Unified World Model, WorldVLA) that jointly capture policy and dynamics through trajectory prediction. Revised Line 080 to explicitly state: "world models act as internal simulators that allow the agent to predict environmental dynamics and plan appropriate actions like policies."
>
> **Provided input representation details:** Added concrete graph structure examples specifying nodes (objects: mug, cabinet, book, plate, dishwasher), edges (relations: inside, close, adjacent, hold, on), and features (object states combined with relations: on, off, open, close) to clarify observation-to-graph transformation process.
>
> **Clarified model size selection:** Explained that baseline methods (LLM-Planner, FLARE, SayCanPay) use Llama-3.2-3B/8B due to requiring stronger reasoning without parameter updates, while TMoW uses only Llama-3.2-1B. This configuration favors baselines with larger capacity, yet TMoW achieves superior performance.

---

### Meta-Review · Area_Chair_YS46 · 2026-01-06

**Summary:**

Summary:
This paper proposes TMoW (Test-time Mixture of World Models), a framework that helps embodied agents adapt to new environments at test time. Instead of using a fixed expert selection, TMoW mixes multiple world models based on scene features at different levels, such as objects and overall context. The routing is updated during testing, and new world models can be added from a few demonstrations without retraining from scratch. Experiments on VirtualHome and ALFWorld show large performance gains over strong baselines, demonstrating better adaptation and generalization to new domains.

Strengths:
1. The paper is well-written.
2. The framework enables dynamic test-time adaptation of world model mixtures through prototype refinement.
3. The distilled model augmentation capability supports continuous expansion of the system's knowledge base.
4. Thorough evaluation of the framework.
Evaluation of the method on real-world scenarios.

Weaknesses (initial reviewer concerns):
1. It is unclear how the router efficiently prevents expert confusion and erroneous activation.
2. The test-time prototype refinement relies on online environmental interaction, and its performance is sensitive to the refinement rate.
3. Lack of discussion on the inference time.
4. There has been prior work addressing dynamic MoE, adaptive MoE, and MoE for continual test-time adaptation (AC comment: the reviewer AbzQ raised this concern, but did not mention specific prior works.)
5. Home environments have low dynamics and thus are unsuitable for evaluation.

**Reviewer Concerns:**

I think the author responses have addressed all reviewer concerns.

**Reviewer Scores:**

Reviewer SnZA, 7Jh2, and fzQq’s initial reviews were already positive. Their initial concerns do not seem to challenge the core contributions of the paper, so the chance of a significant score change could be low.

Reviewer AbzQ had a very low initial score. I think there is a good chance that their score could be raised, given the adequate author responses.

---

### Decision · Program_Chairs · 2026-01-26

Accept (Poster)